# Genome-Wide Identification and Analysis of Chitinase GH18 Gene Family in *Trichoderma longibrachiatum* T6 Strain: Insights into Biocontrol of *Heterodera avenae*

**DOI:** 10.3390/jof11100714

**Published:** 2025-10-01

**Authors:** Cizhong Duan, Jia Liu, Shuwu Zhang, Bingliang Xu

**Affiliations:** 1State Key Laboratory of Aridland Crop Science, Gansu Agricultural University, Lanzhou 730070, China; duancz213@163.com; 2College of Plant Protection, Gansu Agricultural University, Lanzhou 730070, China; jiajia7635724@163.com; 3Gansu Provincial Biocontrol Engineering Laboratory of Crop Diseases and Pests, Lanzhou 730070, China

**Keywords:** biocontrol, glycoside hydrolase 18 chitinase family, expression characteristic

## Abstract

The cereal cyst nematode, *Heterodera avena*, is responsible for substantial economic losses in the global production of wheat, barley, and other cereal crops. Extracellular enzymes, particularly those from the glycoside hydrolase 18 (GH18) family, such as chitinases secreted by *Trichoderma* spp., play a crucial role in nematode control. However, the genome-wide analysis of *Trichoderma longibrachiatum* T6 (T6) *GH18* family genes in controlling of *H. avenae* remains unexplored. Through phylogenetic analysis and bioinformatics tools, we identified and conducted a detailed analysis of 18 *GH18* genes distributed across 13 chromosomes. The analysis encompassed gene structure, evolutionary development, protein characteristics, and gene expression profiles following T6 parasitism on *H. avenae*, as determined by RT-qPCR. Our results indicate that 18 GH18 members in T6 were clustered into three major groups (A, B, and C), which comprise seven subgroups. Each subgroup exhibits highly conserved catalytic domains, motifs, and gene structures, while the *cis*-acting elements demonstrate extensive responsiveness to hormones, stress-related signals, and light. These members are significantly enriched in the chitin catabolic process, extracellular region, and chitinase activity (GO functional enrichment), and they are involved in amino sugar and nucleotide sugar metabolism (KEGG pathway enrichment). Additionally, 13 members formed an interaction network, enhancing chitin degradation efficiency through synergistic effects. Interestingly, 18 members of the *GH18* family genes were expressed after T6 parasitism on *H. avenae* cysts. Notably, *GH18-3* (Group B) and *GH18-16* (Group A) were significantly upregulated, with average increases of 3.21-fold and 3.10-fold, respectively, from 12 to 96 h after parasitism while compared to the control group. Meanwhile, we found that the GH18-3 and GH18-16 proteins exhibit the highest homology with key enzymes responsible for antifungal activity in *T. harzianum*, demonstrating dual biocontrol potential in both antifungal activity and nematode control. Overall, these results indicate that the GH18 family has undergone functional diversification during evolution, with each member assuming specific biological roles in T6 effect on nematodes. This study provides a theoretical foundation for identifying novel nematicidal genes from T6 and cultivating highly efficient biocontrol strains through transgenic engineering, which holds significant practical implications for advancing the biocontrol of plant-parasitic nematodes (PPNs).

## 1. Introduction

Plant-parasitic nematodes (PPNs) are among the most significant soil-borne pathogens that cause diseases in plants, thereby impeding their growth and production [1]. It is estimated that global agricultural production suffers losses up to $215.77 billion annually due to PPNs [2]. *Heterodera avenae* is one of the PPNs that poses a key threat to wheat production [3] and causes substantial economic losses in major wheat-producing regions worldwide (over 40 countries and regions, including North Africa, West Asia, India, China, Australia, and Europe) [4]. In China, losses in wheat production attributed to *H. avenae* ranged from 10% to 35% [5]. Traditional agricultural control measures, such as crop rotation [6] and the use of resistant varieties, can reduce the density of *H. avenae* to a certain extent; however, their implementation is often constrained. This limitation primarily arises from the insufficient economic feasibility associated with long-term crop rotation and the vulnerability of resistant varieties to infection by the second-stage juveniles (J2) of *H. avenae* during the early growth stages [7]. While chemical nematicides are favored for their low cost and rapid effectiveness and have historically served as the primary strategy for managing *H. avenae* [8], their sustainable application is hindered by inherent limitations, including the emergence of nematode resistance, environmental pollution, and ecological hazards [9]. Biological control has the potential to efficiently and specifically manage nematodes, significantly reducing reliance on chemical pesticides and mitigating environmental pollution. The development of biological control agents represents a key direction in promoting the advancement of green agriculture.

*Trichoderma* spp. are widely recognized as effective biological control agents due to their broad-spectrum activity, safety, adaptability, and diverse mechanisms of action [10,11,12,13,14]. Current studies have demonstrated that *T. longibrachiatum* [15], *T. viride* [16], *T. harzianum* [17], *T. koningiopsis* [18], and *T. hamatum* [19] can effectively control PPNs. Zhang et al. [20] found that the *T. longibrachiatum* T6 (T6) strain exhibited intense parasitic and lethal effects on *H. avenae*. One of the main mechanisms for *Trichoderma* spp. against nematodes was mainly relies on various degrading enzymes (chitinase, glucanase, and xylanase, etc.) production and secretion [21], which are beneficial for hyphae to penetrate the eggshells of nematodes or the cuticles of juveniles and adults to colonize, and finally leading to the degradation of the body wall and the death of the nematodes [5]. Among them, chitinase plays a role in the process of parasitism and degradation of the nematode body wall [22]. Our previous studies have demonstrated that T6 exhibits strong parasitic and lethal effects on *H. avenae* by the enhanced extracellular chitinase activity [23]. The chitinase-encoding genes of *Trichoderma* spp. predominantly belong to the *GH18* multigene family [24], which is characterized by a GH18 catalytic domain and several auxiliary domains [25]. Members of this family have been confirmed as key antagonistic factors in the species, including *ThEn-42* from *T. atroviride* [26], *chit33* [27], *chit36* [28], *Chit37* [29], and *Chit42* [30] from *T. harzianum*. Specifically, *ThEn-42* is involved in the biocontrol of *Penicillium digitatum*, *chit33* targets *Rhizoctonia solani*, *chit36* is effective against *Botrytis cinerea*, *Chit37* is directed against *Fusarium oxysporum*, and *Chit42* combats *Sclerotinia sclerotiorum*. In addition, previous studies have primarily concentrated on the species and functions of *GH18* family genes in *Trichoderma* spp., as well as their roles in combating plant pathogens and controlling fungal diseases in plants. However, there is a paucity of information regarding nematodes, particularly PPNs. Studies have demonstrated that the *GH18* family genes *chit42* and *chit33* of *T. harzianum* are significantly upregulated during the parasitism of *Caenorhabditis elegans* eggs, indicating that their chitinases may facilitate the degradation of nematode eggshell [31]; the *GH18* family chitinase gene *T6-Echi18-5* from T6 was markedly upregulated during the infection of *H. avenae* cysts, and that the recombinant protein derived from this gene effectively degrades the eggshell and inhibits egg hatching [32].

The growth and morphological development of fungi necessitate cell wall remodeling, a process that requires the coordinated action of various cell wall-related hydrolases, particularly the GH18 family of chitinases [33,34]. These enzymes enhance nutrient utilization by facilitating endogenous cell wall autolysis and degrading exogenous chitin to meet nutritional demands [35,36,37]. Despite these essential functions, the GH18 family exhibits functional redundancy and significant variation in gene number among *Trichoderma* spp. [38,39]. Even within the same species, *GH18* family genes demonstrate structural and functional domain differentiation. Nevertheless, the precise identification and systematic analysis of *GH18* family genes in T6 have not yet been conducted, which limits a deeper understanding of the functions of its chitinases and their potential in biological control.

In recent years, the overexpression of the key *GH18* gene through genetic engineering has successfully led to the development of engineered strains with enhanced antifungal activity [27], significantly improving biocontrol efficacy against pathogenic fungi and reducing the costs associated with biological disease control. However, research on the key functional information of the GH18 family in T6 remains insufficient, particularly regarding which members are involved in transcriptional regulation and contribute to the nematocidal mechanism during the parasitism of *H. avenae*. This gap in knowledge severely restricts the development of efficient and low-cost biocontrol agents targeting this objective. Therefore, we systematically characterized the *GH18* family genes in T6 through genome-wide identification, including their physicochemical properties, evolutionary relationships, gene structures, and protein domains. Meanwhile, the expression patterns of *GH18* family genes were further characterized within the T6–*H. avenae* parasitism interaction system. This study not only elucidates the molecular mechanisms underlying the nematicidal activity of *GH18* family genes in T6 but also provides significant insights for the development of eco-friendly biological control strategies against *H. avenae*.

## 2. Materials and Methods

### 2.1. Fungal and H. avenae Preparation

The strain of T6 (CGMCC No. 13183) was obtained from the Laboratory of Plant Virology and Molecular Biology, Gansu Agricultural University, Lanzhou, China. Soil samples were collected from the rhizosphere and tillage layer at a depth of 5 to 20 cm using the Z-shaped sampling method. These samples were taken from wheat rhizosphere and the plants exhibiting chlorosis, yellowing, and stunted growth as a result of infection by *H. avenae* in Suzhou City, Anhui Province, China.

### 2.2. Identification and Physicochemical Characterization of the GH18 Family Genes in T6

The genome data of T6 strain and its related annotation files were retrieved from the NCBI database. The genes coding sequences (CDS) of T6 were extracted using the “Gtf/Gff3 Sequences Extract” module of TBtools (v2.034) and then translated into protein sequence. Thereafter, the candidate *GH18* family genes were identified by HMM search, based on the Glyco_hydro_18 (PF00704) profile that was downloaded from Pfam, and the hits were filtered based on their sequence and domain scores, with an E-value cutoff of 1 × 10^−8^. Furthermore, the domain integrity of predicted proteins was further validated using the SMART database (http://smart.embl-heidelberg.de/, E-value threshold <0.0001 (accessed on 24 April 2025)). Additionally, the comprehensive bioinformatics analysis was conducted for identifying the family members, and the parameters of molecular weight, theoretical isoelectric point (pI), instability index, aliphatic index, GRAVY score, subcellular localization, signal peptides, transmembrane topology [40], and phosphorylation sites [41] were analyzed and predicted using the bioinformatics analysis tool that listed in Appendix A.

### 2.3. Genes Chromosomal Location and Ka (Nonsynonymous)/Ks (Synonymous) Analysis

Chromosomal positions and lengths of *the GH18* family genes were obtained from NCBI. Gene locations were visualized using the “Gene Location Visualize from GTF/GFF file” function within TBtools [42]. *GH18* genes were mapped onto the T6 genome and renamed according to their chromosomal distribution. The selection and evolutionary pressure values of the *GH18* family in T6, *T. harzianum*, *T. virens*, and *T. reesei* were calculated using the Ka/Ks Calculator (v2.0) [43,44].

### 2.4. Proteins Domain, Genes Structure, and Conserved Motif Analysis

The gene structure, specifically the exon-intron organization, was visualized using TBtools, based on genome annotation files [45]. The domain composition of GH18 family proteins was identified using SMART. Domain features were further validated through the PROSITE database [46] (https://prosite.expasy.org/ (accessed on 25 April 2025)). Conserved motifs were predicted using the MEME Suite [47] (http://memesuite.org/tools/meme (accessed on 27 April 2025)). The motif data were subsequently integrated and visualized using the “Gene Structure View (Advances)” function in TBtools.

### 2.5. Phylogenetic Analysis

The GH18 family protein sequences of *T. harzianum*, *T. virens*, and *T. reesei* were retrieved from the NCBI and Phytozome databases. The protein sequences were aligned utilizing ClustalX (v2.1). A neighbor-joining phylogenetic tree was constructed using MEGA11 (v11.0.13), incorporating 1000 bootstrap replicates and default parameters. The resulting tree was visualized and annotated with the iTOL server [48].

### 2.6. Cis-Acting Element Analysis

Promoter sequences, specifically those located 2000 base pairs upstream of the translation start site of T6 *GH18* family genes, were extracted from the genome database using the “Gtf/Gff3 Sequences Extract” and “Fasta Extract (Recommended)” functions within TBtools. Subsequently, these sequences were submitted to the PlantCARE database (http://bioinformatics.psb.ugent.be/webtools/plantcare/html/ (accessed on 28 April 2025)) for the analysis of *cis*-acting elements [49]. The distribution of *cis*-elements was visualized using TBtools [42].

### 2.7. GO and KEGG Enrichment Analysis

To functionally annotate the *GH18* family genes, a Gene Ontology (GO) comparative analysis was conducted using eggNOG-mapper (http://eggnog5.embl.de/ (accessed on 28 April 2025)). Visualization of the results was performed using WEGO (v2.0). The KEGG pathway annotation file was obtained from the KEGG database (https://www.kegg.jp/kegg/pathway.html (accessed on 29 April 2025)). KEGG pathway enrichment analysis was conducted utilizing the cluster Profiler package (v3.21), and the results were visualized using the ggplot2 plotting system (v3.21).

### 2.8. GH18 Proteins Structural Prediction and Interaction Network Analysis

The secondary structure of GH18 proteins was predicted using NPS@SOPMA (v2.0). Tertiary structure was modeled with AlphaFold via UniProt (https://www.uniprot.org/ (accessed on 29 April 2025)), which generated high-confidence three-dimensional models [50,51]. Additionally, protein–protein interaction networks were predicted using STRING (https://cn.string-db.org/, accessed on 30 April 2025).

### 2.9. Gene Expression Characteristics Analysis

To analyze the *GH18* family genes expression characteristics after the T6 parasitism on the *H. avenae* cysts, the methods and samples preparation were conducted according to Wang et al. [52] with minimal modification. The T6 mycelia were collected at 12, 24, 48, 72, and 96 h after T6 parasitized on the surface of cysts in the treatment and control groups, respectively. The mycelia samples were immediately frozen in liquid nitrogen and stored at −80 °C until for using. Total RNA from each sample was converted to cDNA using the RevertAid First Strand cDNA Synthesis Kit (Invitrogen, Waltham, MA, USA), and then the real-time quantitative PCR (RT-qPCR) was performed using the TB Green^®^ Premix Ex Taq^™^ II kit (Takara, Dalian, China) with Actin (accession number XM_024923451.1) as the internal reference gene. Appendix A lists the specific primers for *GH18* family genes and the internal reference gene used for RT-qPCR analysis. RT-qPCR was conducted on the Thermo QuantStudio^™^ 5 thermal cycler (Thermo Fisher Scientific, Waltham, MA, USA). The 2^−ΔΔCT^ method was then employed to calculate the relative gene expression. Three biological replicates were conducted for each amplification reaction.

## 3. Results

### 3.1. Genome-Wide Identification and Physicochemical Properties Analysis of GH18 Family Genes in T6

Based on the genomic data of T6, a total of 20 protein sequences containing the glycoside hydrolase family 18 catalytic domain (PF00704) were identified using a Hidden Markov Model (HMM) with an e-value threshold of 1 × 10^−8^. By integrating genome annotation data and utilizing TBtools software, the domain integrity of the predicted proteins was further validated through comparisons with the GH18 sequences of *T. reesei* and the SMART database. Ultimately, 18 members were confirmed to belong to the *GH18* family genes in T6. These genes were designated as *GH18-1* to *GH18-18* based on their chromosomal localization. The ORF of the *GH18* family members varied in length from 966 to 4512 bp, encoding peptides ranging from 321 to 1503 amino acids (Table 1). This length variation was primarily attributed to the presence of a C-terminal serine/threonine kinase domain. Physicochemical property analysis revealed that the GH18 proteins exhibited molecular weights from 34.53 to 159.28 kDa, with theoretical isoelectric points (pI) ranging from 4.20 to 6.20. According to the instability index, 10 members (index < 40) were predicted to be stable proteins, while the remaining 8 members (index 40.03 to 46.59) were classified as unstable. The aliphatic index varied from 64.18 to 90.38, indicating significant differences in thermal stability among the family members. Except for GH18-6, which displayed a positive grand average of hydropathicity (GRAVY) value, all other proteins had negative GRAVY values (−0.592 to −0.031), categorizing them as typical hydrophilic proteins. Transmembrane domain prediction indicated that none of the GH18 proteins possessed transmembrane helices. Given that signal peptides are crucial for determining secretion capacity in chitinases, further analysis identified 11 members (GH18-3, GH18-4, GH18-6, GH18-8, GH18-10, GH18-11, GH18-12, GH18-14, GH18-15, GH18-16, GH18-17) that contained signal peptides, while the remaining 7 members lacked them. Subcellular localization predictions indicated that all GH18 proteins, except for GH18-1 and GH18-13, which localized to mitochondria, were extracellular (Table 1). Phosphorylation site prediction (Figure 1) confirmed the presence of multiple potential phosphorylation sites across all GH18 proteins.

### 3.2. Chromosomal Localization, Ka (Nonsynonymous)/Ks (Synonymous) Analysis of GH18 Family Genes in T6

The genomic distribution analysis showed that the 18 *GH18* genes were widely distributed across 13 chromosomal segments (Figure 2). The number of *GH18* family genes on each chromosome was not correlated with chromosome size. Specifically, *GH18-1* and *GH18-2* were located on chromosome KZ679126; *GH18-3* and *GH18-4* on KZ679128; *GH18-5* and *GH18-6* on KZ679129; *GH18-12* and *GH18-13* on KZ679137; *GH18-17* and *GH18-18* on KZ679151. Gene density exhibits no correlation with chromosome size. In T6, among the 18 members of the GH18 family, three gene duplication events were identified, with Ka/Ks ratios ranging from 0.2453 to 0.2790, all of which are less than 1 (Figure 3). Similarly, there were 5, 12, and 19 gene duplication events in *T. reesei*, *T. harzianum*, and *T. virens*, respectively, with their Ka/Ks ratios also remaining below 1.

### 3.3. Phylogenetic Analysis of GH18 Family Members in T6

To further understand the functional diversity and evolutionary relationships of the T6 *GH18* family genes across species, GH18 protein sequences from *T. harzianum*, *T. virens*, and *T. reesei* were compared, and a phylogenetic tree was constructed (Figure 4). Based on sequence similarity of the GH18 catalytic domains, the 18 T6 GH18 proteins clustered with 58 GH18 proteins from the other three *Trichoderma* spp. into three major groups (A, B, and C) and eight distinct subgroups (A-II, A-IV, A-V; B-I, B-II, B-V; C-I, C-II). This classification was consistent with domain conservation analyses.

### 3.4. Conserved Motif, Domain, and Gene Structure Analysis of GH18 Family Members in T6

We conducted an analysis of 18 members of the GH18 family, which are divided into three subgroups (Figure 5A). Pfam domain analysis indicated that all GH18 proteins contained the Glyco_18 domain, which was characteristic of GH18 chitinases (Figure 5B). Remarkably, Group C possessed a unique LysM domain. To characterize the GH18 family proteins, the conserved motifs were analyzed using MEME. Twenty motifs ranging from 14 to 48 amino acids were identified across the 18 GH18 sequences (Figure 6, Table 2). In group A, motifs 6, 5, 2, 8, 4, and 1 formed a conserved domain; in group B, motifs 5 and 2 were consistently present; group C proteins shared motifs 3, 7, 10, 6, 1, 9, and 8, constituting the conserved GH18 domain. Notably, motifs 5 and 2 appeared in nearly all GH18 sequences except GH18-2. Some motifs (6, 5, 2, 8, 4, and 1) were exclusive to genes within the same group, suggesting similar functional roles (Figure 5C). The motif distribution patterns correlated well with phylogenetic relationships. Gene structure analysis revealed that the 18 *GH18* genes contained 1 to 12 exons, with most harboring 3 to 4 (Figure 5D).

### 3.5. Cis-Acting Element Analysis of GH18 Family Genes in T6

*Cis-acting* element analysis of the *GH18* gene promoters revealed 12 common types of elements that are widely distributed among family members (Figure 7A). These include auxin-responsive, regulatory element anaerobic, MeJA-responsive, SA-responsive, cell cycle regulation, MYB binding site, light-responsive, gibberellin-responsive, zein metabolism regulation, abscisic acid-responsive, low-temperature-responsive, and defense- and stress-responsive types. Phytohormone-responsive elements (including auxin-responsive, MeJA-responsive, SA-responsive, gibberellin-responsive, and abscisic acid-responsive elements) were the most prevalent and phylogenetically conserved regulatory features, identified in the promoter regions of 16 *GH18* gene members, with the exception of *GH18-17* and *GH18-18*. Stress-responsive elements (encompassing regulatory, anaerobic, low-temperature-responsive, defense-responsive, and stress-responsive elements) were presented in 12 members. Light-responsive elements occurred in 17 members, with *GH18-3* containing the highest number (11), while *GH18-17* lacked them entirely. Among developmental and metabolic process regulatory elements, cell cycle regulation elements were exclusive to *GH18-1*, *GH18-5*, and *GH18-11*, whereas zein metabolism regulation elements were detected in *GH18-1*, *GH18-2*, *GH18-3*, *GH18-4*, *GH18-5*, *GH18-7*, *GH18-11*, *GH18-14*, *GH18-15,* and *GH18-16*. MYB transcription factor binding sites were found in 12 members, with *GH18-6*, *GH18-12*, *GH18-14*, *GH18-15*, *GH18-17,* and *GH18-18* showing their absence (Figure 7B).

### 3.6. Prediction of GH18 Interacting Proteins and Analysis of Protein Secondary and Tertiary Structures in T6

The GH18 proteins are predicted to consist primarily of α-helix (14.50–36.64%), extended strand (10.40–17.93%), and random coil (47.07–71.59%), with no β-turns present (Table 3). The α-helix and random coil are the most abundant structural elements among all family members, achieving an evolutionary balance between conformational flexibility and catalytic rigidity. AlphaFold models indicate that GH18 proteins adopt the TIM8-barrel (also known as TIM barrel) folding pattern, which is characteristic of GH18 chitinases and is a common β-folded barrel structure composed of eight parallel β-folded chains. Within each subclade, the 3D structures are highly conserved; however, clear structural differences are evident between subclades (Figure 8).

The protein–protein interaction network serves as a vital resource for predicting functional orthologs within homologous gene clusters, thereby offering a significant foundation for elucidating gene interactions and regulatory networks. A considerable number of interactions among the 13 GH18 family proteins indicated that they may collectively establish an interaction network that enhances the efficiency of chitin degradation (Figure 9).

### 3.7. GO and KEGG Enrichment Analysis of GH18 Family Genes in T6

To explore the functional annotation of the T6 *GH18* family genes, Eggnog was employed to annotate the potential functions of all identified *GH18* genes. A total of 36, 10, and 3 GO terms were enriched in biological processes, cellular components, and molecular functions, respectively (Figure 10A). Regarding the biological processes, *GH18* family genes are primarily enriched in the chitin metabolic process, glucosamine-containing compound metabolic process, carbohydrate derivative catabolic process, aminoglycan metabolic process, and amino sugar metabolic process. In terms of cellular components, *GH18* family genes are predominantly enriched in the extracellular region. Furthermore, concerning molecular functions, *GH18* family genes are mainly enriched in chitinase activity, hydrolase activity (acting on glycosyl bonds), and carbohydrate derivative binding. KEGG pathway enrichment analysis indicated that 15 genes in the T6 *GH18* family (*GH18-1*, *GH18-3*, *GH18-4*, *GH18-5*, *GH18-7*, *GH18-8*, *GH18-9*, *GH18-10*, *GH18-11*, *GH18-12*, *GH18-13*, *GH18-14*, *GH18-15*, *GH18-16*, *GH18-17*) were significantly enriched in the amino sugar and nucleotide sugar metabolism pathway (Figure 10B).

### 3.8. Relative Expression Level of GH18 Family Genes in T6 After Inoculation with Cysts

To investigate the transcriptional responses of *GH18* family genes during T6 parasitism on *H. avenae* cysts, RT-qPCR was performed to assess the relative expression levels of 18 *GH18* genes in T6 at various time points (Figure 11).

At 12 h after the T6 parasitized on the surface of the cysts, the expression levels of *GH18-1*, *GH18-3*, *GH18-4*, *GH18-8*, *GH18-10*, *GH18-11*, *GH18-15*, and *GH18-16* were significantly upregulated compared to the control group (*p* < 0.05). At 24 h, significant upregulation was noted in *GH18-1*, *GH18-3*, *GH18-4*, *GH18-6*, *GH18-8*, *GH18-10*, *GH18-14*, *GH18-15* and *GH18-16* (*p* < 0.05). At 48 h, *GH18-3*, *GH18-6*, *GH18-7*, *GH18-11*, *GH18-13* and *GH18-16* exhibited significant elevation (*p* < 0.05). At 72 h, most genes showed significant upregulation (*p* < 0.05), while at 96 h, only *GH18-4* remained significantly upregulated. The proportion and types of significantly upregulated genes varied across infection stages, indicating functional differentiation within the *GH18* family genes. Notably, the expression levels of the *GH18-3* (Group B) and *GH18-16* (Group A) at 12, 24, 48, and 72 h were significantly higher compared to the control group. The expression level of the *GH18-3* gene increased by 2.03, 6.67, 4.63, 1.70, and 1.18-fold at 12, 24, 48, 72, and 96 h, compared to the control group. Similarly, the expression level of the *GH18-16* gene was increased by 2.00, 4.29, 4.70, 2.68, and 1.81-fold at the same time points. Combined with genomic information analysis and quantitative data, it appears that *GH18-3* and *GH18-16* play significant roles in T6 parasitism of *H. avenae* cysts.

## 4. Discussion

Chitinases constitute a group of highly specific enzymes that catalyze the hydrolysis of the β-1,4 glycosidic bond in chitin. These enzymes are widely distributed in nature [53]. Within the genus *Trichoderma*, the GH18 family of chitinases constitute a diverse multigene family that plays a critical role in essential biological processes, including growth, nutrient acquisition, interspecies interactions, and defense [54,55]. The fungal *GH18* gene family exhibits dynamic evolutionary features [56], with interspecies variation in gene numbers ranging from 10 to 30 members [57,58,59]. In this study, a genome-wide analysis identified 18 *GH18* family genes in T6, consistent with *T. reesei* [39]. We also analyzed the physicochemical properties of the T6 GH18 family members. Detailed characterization revealed that 88% (16/18) of these GH18 chitinases exhibited acidic isoelectric points (pI 4.20–6.20), aligning with previous findings that most GH18 chitinases in *T. reesei* are acidic enzymes which catalytic activity depends on acidic amino acid residues [60]. The GRAVY values and elevated aliphatic indices collectively corroborate their inherent hydrophilicity and thermodynamic stability. Subcellular localization prediction indicated that, with the exception of two proteins in the A-IV subgroup predicted to localize to mitochondria, all others were extracellular. Within the GH18 family of T6, 11 members possess a signal peptide, whereas 15 members in *T. reesei* exhibit this characteristic as well, indicating that these proteins are likely secreted [39]. The lack of a typical N-terminal signal peptide in the remaining members suggests that they may be secreted through non-classical pathways [61]. All GH18 proteins contain multiple phosphorylation sites, indicating that post-translational modifications regulate their biological functions. In this study, the *GH18* genes demonstrated a conserved evolutionary arrangement across chromosomes. This distribution pattern is consistent with the chromosomal organization observed in the *GH18* families across most *Trichoderma* spp., likely underpinning their functional diversification and transcriptional regulation [62]. *Ka* (nonsynonymous substitution rate) and *Ks* (synonymous substitution rate) represent the frequencies of nonsynonymous and synonymous substitutions in gene sequences, respectively. The Ka/Ks ratio effectively quantifies the selective pressures acting on genes during evolution by comparing the relative rates of these two types of substitutions [63]. The Ka/Ks analysis revealed that the *GH18* genes in T6 underwent three gene duplications and experienced purifying selection (Ka/Ks < 1), exhibiting an evolutionary pattern most similar to that of the *GH18* genes in *T. reesei*. This purifying selection indicates that the GH18 protein have maintained functional conservation, thereby preserving their core functions [64].

Phylogenetic analysis clustered the 18 T6 GH18 proteins into three groups (A, B, and C), which were further subdivided into seven subgroups. This classification is consistent with previous studies on the species [25,39]. Studies have demonstrated that both Group A and Group B of the GH18 family in fungi have experienced non-random evolutionary processes. Notably, the C-I and C-II subgroups within Group C appear to be the primary contributors to this non-random expansion. In comparison to other subgroups in the GH18 family, the C-I and C-II subgroups exhibit significant interspecies sequence variability. This characteristic can be attributed to the effects of diversifying selection and their evolutionary adaptation to variations in the cell wall composition of antagonistic species [39]. Similarly, the majority of C-I and C-II in the fungi GH18 family demonstrate significant interspecies sequence variability, which is driven by positive selection and diversification evolution [56]. Group A members (subgroups A-II/A IV/A-V) possess a single catalytic domain devoid of carbohydrate-binding modules (CBMs), which is homologous to the majority of chitinases found in filamentous fungi. These members are primarily associated with cell wall degradation [62] and mycoparasitism [65]. Group B proteins (subgroups B-I/B-II/B-V) exhibit significant variations in size and structure. These enzymes typically contain C-terminal carbohydrate-binding modules (CBMs) or serine/threonine-rich domains. They play a crucial role in cell wall remodeling processes, as evidenced in *Aspergillus nidulans* [66], and are induced by chitin or specific carbon source nutrients derived from the host [67]. Group C (subgroup C-II) contains a chitin-binding domain and one to two lysM-motif short peptide domains, which facilitate the binding of peptidoglycan and its structurally related molecules [64]. This group represents a novel subgroup of fungal chitinases, which domains are entirely distinct from those of Groups A and B, indicating that these enzymes possess unique characteristics in the chitin degradation process. Additionally, members of the *GH18* gene group exhibit similar conserved domains and motif distributions, indicating functional similarities among the proteins within the same group. Gene structure analysis reveals that the genes in Group A contain a greater number of introns compared to those in Groups B and C. This finding suggests that Group A genes may enhance the evolution of biological functions through mechanisms such as recombination and increased complexity during the evolutionary process [68].

The identification of 12 types of *cis*-acting elements in the promoters of the *GH18* family genes in T6 reveals significant characteristics of this gene family regarding environmental response plasticity and functional evolutionary specialization. The widespread distribution of hormone-responsive elements in the promoter regions of the *GH18* family genes suggests that these genes may be finely regulated by multiple hormones. Additionally, the presence of environmental stress-responsive elements in the promoter regions indicates the adaptability of T6 to various adverse conditions. Light-responsive elements regulate spore formation and circadian rhythm-related genes, such as the *blr-1/2* photoreceptor genes, which are key regulatory elements in photoconidiation and mycelial growth in *T. atroviride* and induce spore formation in *T. atroviride* [69]. The specific distribution of developmental and cellular process regulatory elements within the promoter regions of *GH18* family genes suggest their potential roles in the growth and development of T6. The characteristics of these *cis*-acting elements provide important molecular insights for a deeper understanding of the ecological adaptability and biocontrol functions of T6.

The GO functional enrichment indicates that GH18-3, GH18-7, GH18-14, GH18-16, and GH18-17 are primarily enriched in processes related to chitin catabolism, which includes the biochemical reactions and pathways involved in the degradation of chitin. Regarding cellular components, these genes are predominantly enriched in the extracellular region, with their molecular function annotated as chitinase activity, specifically hydrolyzing GlcNAc polymers found in chitin and chitooligo-saccharides. This is similar to the GO functional annotation of the GH18 family T6-Echi18-5 protein in T6, as previously reported by Shen et al., which plays a significant role in the degradation of *H. avenae* eggs by T6 [33]. This suggests that these genes provide a carbon source for the parasitic growth of T6 by participating in the degradation of complex carbohydrates, such as chitin, and are involved in cell wall synthesis and the maintenance of energy metabolism [70]. Furthermore, this study’s KEGG enrichment analysis revealed that 15 genes were enriched in the amino sugar and nucleotide sugar metabolism pathway, which is one of the core metabolic networks essential for fungal growth and development. Previous studies have demonstrated that these pathway significantly influences fungal morphological differentiation, nutrient acquisition, and environmental responses by regulating processes such as cell wall chitin synthesis [71] and carbon source utilization [72].

During the parasitism of T6 on *H. avenae* cysts, the transcriptional level of the *GH18* family genes *GH18-3* and *GH18-16* exhibited a significant upregulation. Interestingly, homology modeling revealed a significant structural similarity between T6 GH18-3 and *T. harzianum* Chit33, a chitinase recognized for its antifungal activity against *R. solani* [25]. In addition, the expression of *Chit33* (*T. harzianum*) is induced by chitin, suggesting that it may degrade exogenous chitin for nutritional purposes [73]. GH18-16 (Group A) demonstrates a high degree of structural homology with Chit42 from *T. harzianum*. Compared to the wild type, the chitinase activity in *Chit42*-overexpressing transformants is significantly enhanced, resulting in a 4.98-fold increase in antifungal activity against *S. sclerotiorum* [30]. In the presence of pathogenic fungal (*R. solani*), which contains a high proportion of chitin, the GH18 family group A genes *AO-190* and *AO-801* of the nematode-trapping fungus *Arthrobotrys oligospora* exhibit enhanced expression. This observation indicates their potential involvement in biological control and mycoparasitism processes [74]. It is noteworthy that the nematode cyst wall is primarily composed of chitin (approximately 40%) and protein fibers, which together form a complex microfibrillar structure [75]. These findings collectively suggest that *GH18-3* and *GH18-16* may possess the capability to degrade the cell walls of pathogenic fungi and the cyst walls of nematodes, thereby endowing them with potential dual functions of antimicrobial and nematicidal activities. Furthermore, during various stages of infection, the significantly upregulated chitinase genes exhibit variations in both proportion and type. This observation indicates that the *GH18* gene family has undergone functional differentiation during evolution, implying that specific types of chitinase genes may perform distinct biological functions in the parasitic infect of *Trichoderma* spp. on nematodes. This discovery closely aligns with the transcriptional expression profiles of the *GH18* family genes *chit18-2* (Group A), *chi18-4* (Group A), *chi18-13* (Group A), *chi18-3* (Group B), and *chi18-10* (Group C) in *T. atroviride*, which expression is influenced by varying growth conditions and carbon sources [40]. This suggests that the functions of GH18 family chitinases in *Trichoderma* spp. are diverse; they are not mutually substitutable but rather fulfill specific roles within the organism. Similarly, the *GH18* family genes in *A. oligospora* exhibit differential expression at various growth stages and perform multiple functions throughout growth, differentiation, and infection [74]. Interestingly, the STRING protein–protein interaction network analysis revealed that the *GH18* family genes do not function in isolation; rather, the 13 identified GH18 family proteins form a tightly interconnected network. It is speculated that in T6 exists a multi-enzyme complex system specifically targeting chitin degradation, which significantly enhances substrate degradation efficiency through cascade reactions. In conclusion, *GH18-3* and *GH18-16* are important candidate genes in T6 for further research and detailed characterization to elucidate their roles in the infection of *H. avenae*.

## 5. Conclusions

Bioinformatic analysis revealed the evolutionary relationships and conserved characteristics of the *GH18* gene family in T6, while expression quantification analysis elucidated the regulatory mechanisms of these genes during the parasitism of *H. avenae* cysts. Crucially, *GH18-3* and *GH18-16* were identified as key genes interacting with nematodes, providing essential targets for in-depth exploration of the molecular mechanisms by which *Trichoderma* spp. chitinases parasitism on *H. avenae*. The findings of this study not only enhance the fundamental theoretical research on chitinase-based biopesticides but also provide an external impetus for the genetic engineering modification of strains, thereby promoting the advancement of *Trichoderma* field control technologies. Furthermore, this research offers new insights into the development of environmentally friendly strategies for controlling PPNs.

## Figures and Tables

**Figure 1 jof-11-00714-f001:**
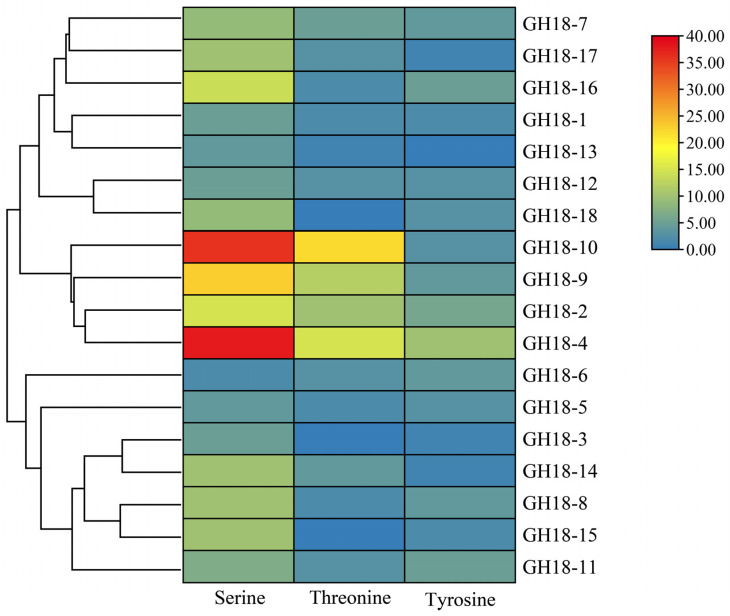
The phosphorylation sites in the GH18 family proteins of T6.

**Figure 2 jof-11-00714-f002:**
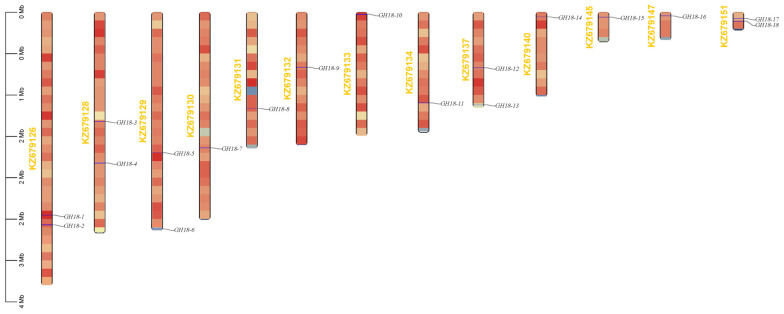
Chromosomal localization of the *GH18* family genes in T6. Chromosome positioning was based on the physical location of the 18 T6 *GH18s*. Genes names are indicated in black. The scale bar is on the left.

**Figure 3 jof-11-00714-f003:**
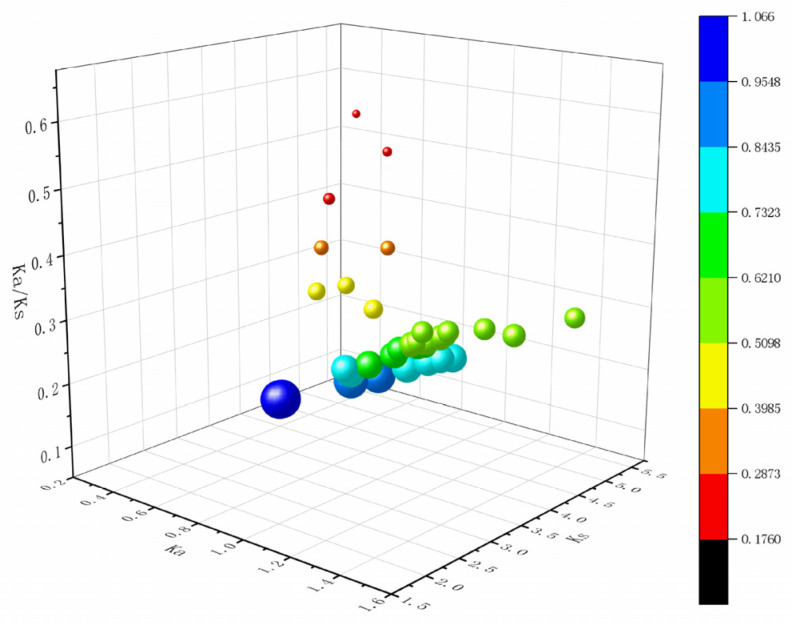
Evolutionary selection pressure analysis of *GH18* homologous gene pairs. The X-axis represents the Ka value, the Y-axis represents the Ks value, and the Z-axis represents the ratio of Ka to Ks. The color scale represents the fold change normalized by log2-transformation of the data.

**Figure 4 jof-11-00714-f004:**
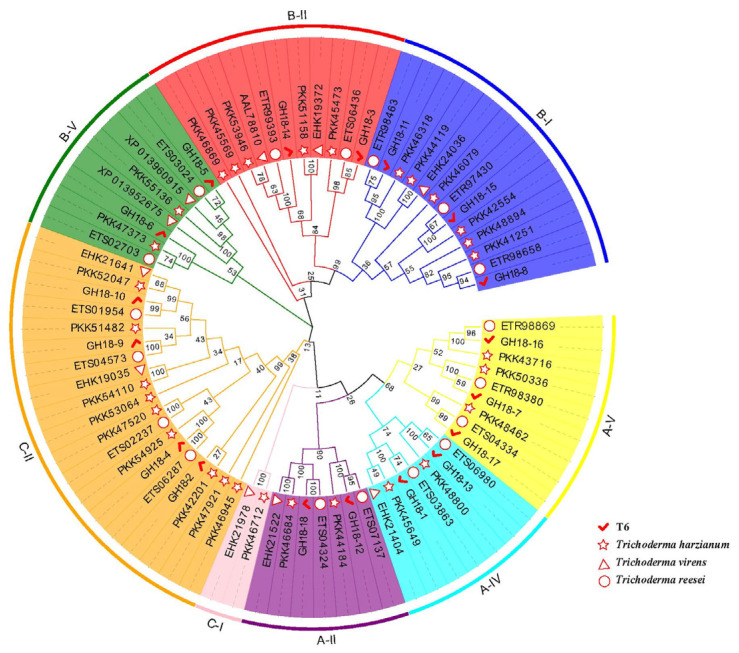
The phylogenetic tree was constructed using the neighbor-joining (NJ) method with MEGA11 (v11.0.13), based on the amino acid sequences of T6, *T. harzianum*, *T. virens*, and *T. reesei*. Putative GH18 proteins were identified in these species based on the results of an HMM search for PF00704. Furthermore, these proteins were classified into three major groups (A, B, and C) and eight subgroups (A-II, A-IV, A-V; B-I, B-II, B-V; C-I; C-II).

**Figure 5 jof-11-00714-f005:**
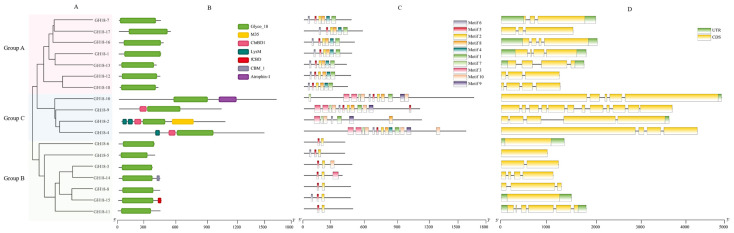
Analysis of the gene structure of 18 *GH18* family members. (**A**) Phylogenetic tree. Different colors (Red, blue, and green) indicate different groups. (**B**) Domain defined using the PFAM database. (**C**) Motif analysis. The length and different colors of boxes denote motif length and different motifs, respectively. (**D**) Gene structure. CDS denotes exons.

**Figure 6 jof-11-00714-f006:**
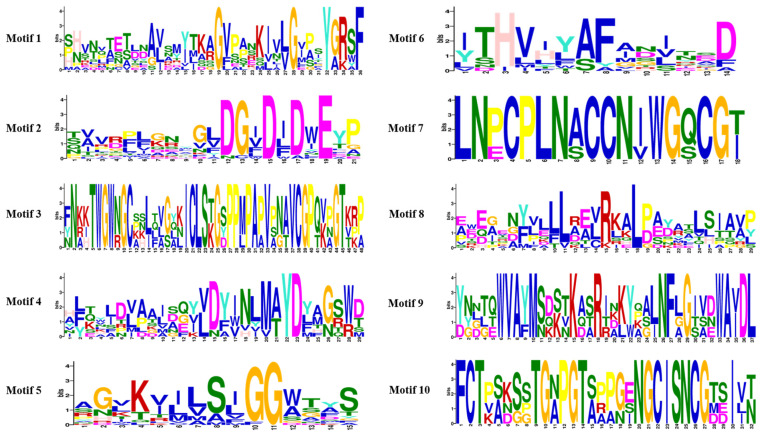
Motif composition of the GH18 family proteins of T6. The amino acid sequences of various conserved motifs are depicted as stacked letters at each position. The total height of each stack signifies the relative amino acid information content for each letter at the motif position, quantifies in bits. The height of each letter within the stack is determined by multiplying the probability of that letter at the respective position by the overall information content of the stack. The X-axis and Y-axis represent the width and the number of bits per letter, respectively, thereby indicating the relative amino acid information content at each motif position.

**Figure 7 jof-11-00714-f007:**
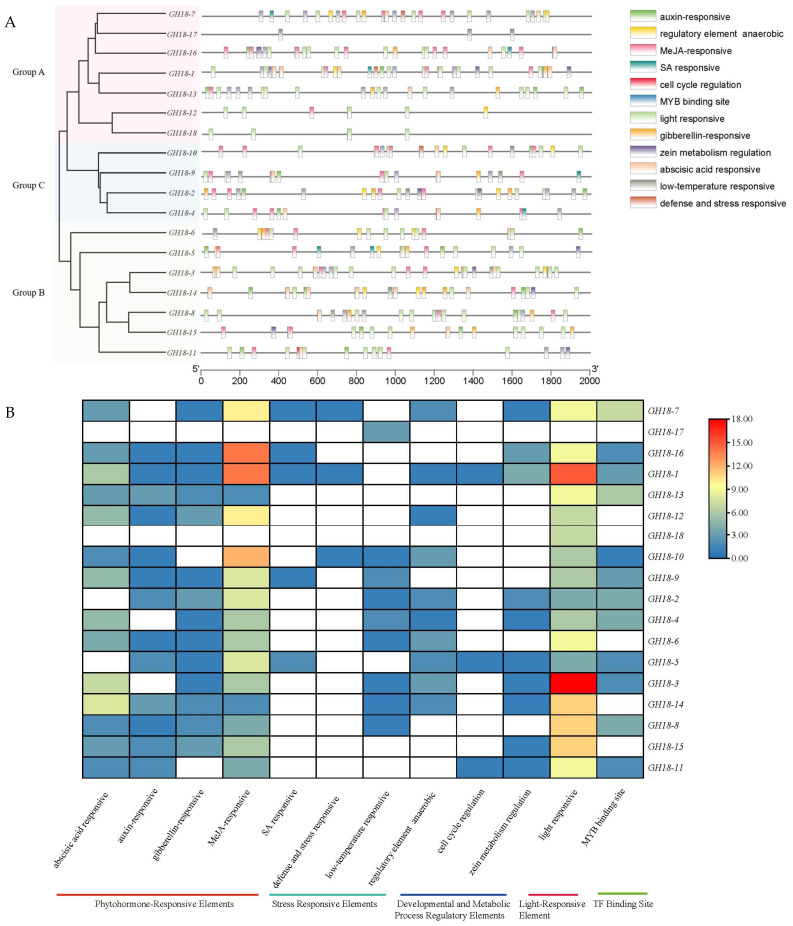
*Cis*-acting elements analysis of T6 *GH18* family genes. (**A**) Analysis of *cis*-acting elements of the *GH18* family genes in T6. Different colored wedges are utilized to represent various *cis*-acting elements. The length and position of each *GH18* gene are accurately mapped to scale, with the scale bar indicating the length of the DNA sequence. (**B**) The number of *cis*-acting elements in T6 *GH18* family genes. The varying colors and numbers within the grid represent the quantities of distinct *cis*-acting elements presented in these *GH18* genes.

**Figure 8 jof-11-00714-f008:**
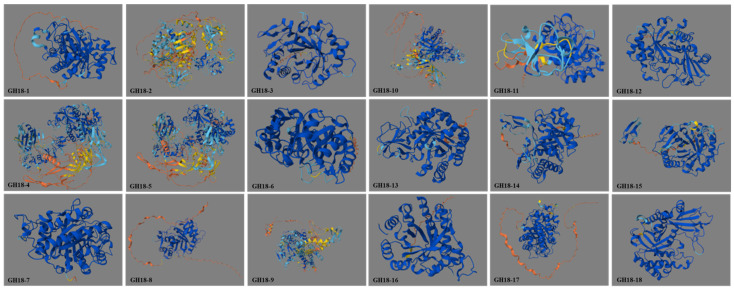
Predicted tertiary structure of T6 GH18 family proteins.

**Figure 9 jof-11-00714-f009:**
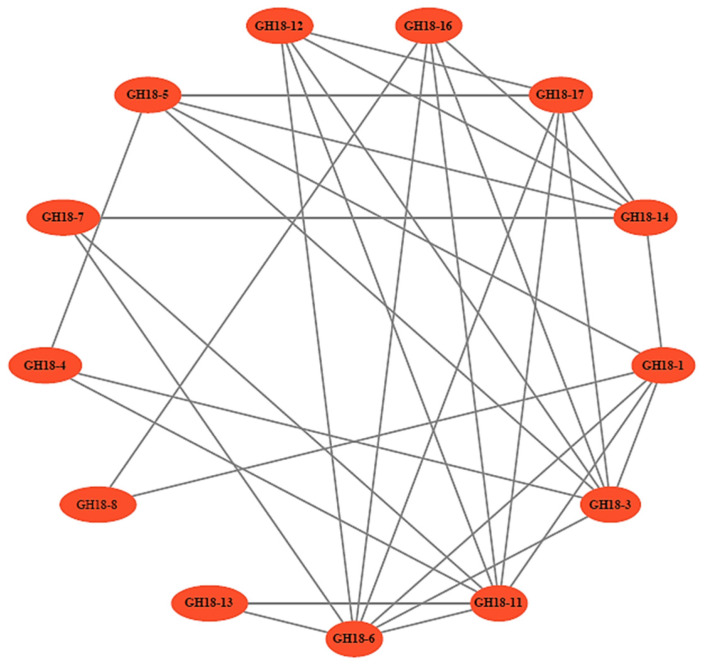
The protein–protein interaction prediction network of T6 GH18 family proteins. Each node represents a protein, and each edge represents an interaction.

**Figure 10 jof-11-00714-f010:**
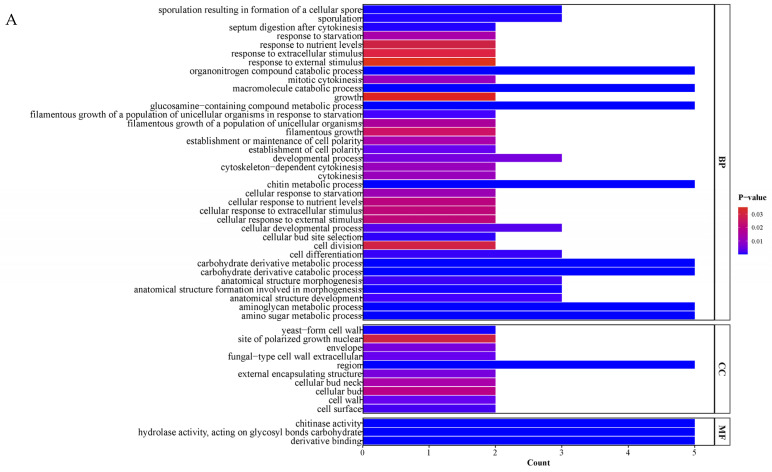
GO and KEGG enrichment analysis of *T6 GH18* family genes. (**A**) Bar chart of GO enrichment analysis for 18 *GH18* genes. (**B**) Scatter plot of KEGG enrichment analysis for 18 *GH18* genes.

**Figure 11 jof-11-00714-f011:**
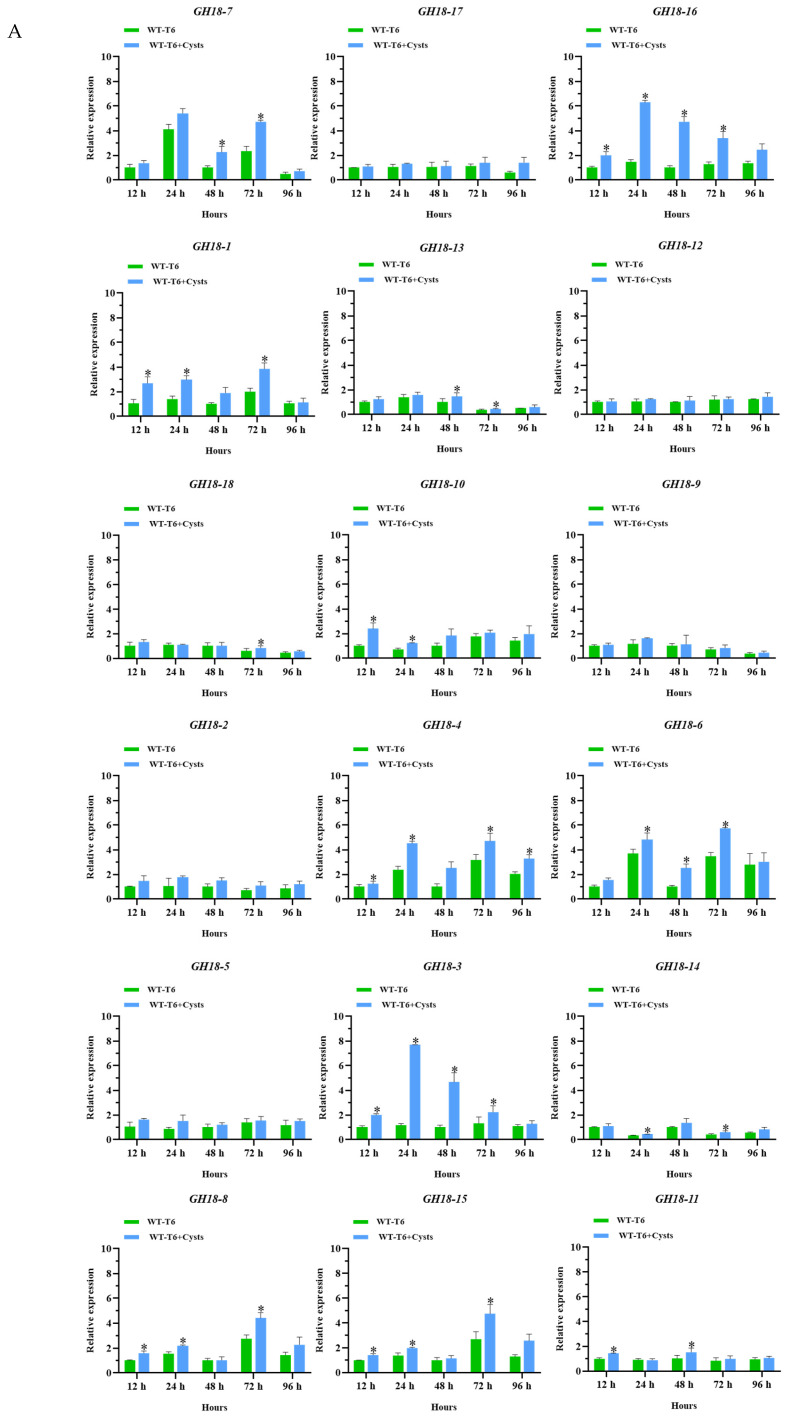
The relative expression levels of the *GH18* family genes at different time points after T6 mycelia parasitized on the surface of the *H. avenae* cysts. *Actin* served as the internal reference gene, with the expression level of each gene member at 12 h used as the control. (**A**) Detailed expression patterns, the relative expression data were analyzed using paired sample *t-tests* to identify significant differences, with * indicating a significant difference at the *p < 0.05* level. The error bars represent standard error calculated from three replicates. (**B**) The expression levels are illustrated in the form of a heat map.

**Table 1 jof-11-00714-t001:** Physical and chemical properties of the T6 GH18 members.

Genes	Genes ID	Genes Locus	Amino Acid (aa)	ORF (bp)	Molecular Weight (kDa)	Theoretical pI	Instability Index	Aliphatic Index	GRAVY Score	Transmembrane Domain	Signal Peptides	Subcellular Localization
*GH18-1*	M440DRAFT_1367966	KZ679126	401	1206	44.25	5.96	46.59	72.02	−0.458	No	No	Mitochondrion
*GH18-2*	M440DRAFT_1459816	KZ679126	990	2973	106.44	5.44	37.22	69.76	−0.304	No	No	extracellular
*GH18-3*	M440DRAFT_1436691	KZ679128	321	966	34.53	4.20	40.03	67.04	−0.213	No	Yes	extracellular
*GH18-4*	M440DRAFT_1326396	KZ679128	1386	4161	147.06	4.69	35.20	69.07	−0.260	No	Yes	extracellular
*GH18-5*	M440DRAFT_1390232	KZ679129	342	1029	38.24	5.33	39.97	90.38	−0.234	No	No	extracellular
*GH18-6*	M440DRAFT_1400047	KZ679129	343	1032	36.22	4.64	27.72	85.13	0.093	No	Yes	extracellular
*GH18-7*	M440DRAFT_1400627	KZ679130	397	1194	44.26	5.11	30.76	64.18	−0.592	No	No	extracellular
*GH18-8*	M440DRAFT_1239474	KZ679131	390	1173	40.68	4.55	45.97	71.38	−0.142	No	Yes	extracellular
*GH18-9*	M440DRAFT_66100	KZ679132	975	2928	107.32	5.64	42.07	75.34	−0.296	No	No	extracellular
*GH18-10*	M440DRAFT_1470525	KZ679133	1503	4512	159.28	6.01	33.86	67.86	−0.338	No	Yes	extracellular
*GH18-11*	M440DRAFT_1392779	KZ679134	403	1212	41.22	4.60	28.17	71.59	−0.031	No	Yes	extracellular
*GH18-12*	M440DRAFT_1338469	KZ679137	393	1182	43.99	4.78	33.72	89.11	−0.204	No	Yes	extracellular
*GH18-13*	M440DRAFT_1058965	KZ679137	357	1074	38.57	6.20	40.29	83.39	−0.087	No	No	Mitochondrion
*GH18-14*	M440DRAFT_59808	KZ679140	392	1179	41.45	4.21	42.60	65.99	−0.174	No	Yes	extracellular
*GH18-15*	M440DRAFT_1434173	KZ679145	410	1233	42.47	4.48	42.28	69.78	−0.134	No	Yes	extracellular
*GH18-16*	M440DRAFT_1406318	KZ67914	424	1275	45.89	5.29	25.35	80.42	−0.204	No	Yes	extracellular
*GH18-17*	M440DRAFT_1344625	KZ679151	492	1479	54.31	5.37	44.63	77.74	−0.383	No	Yes	extracellular
*GH18-18*	M440DRAFT_1395841	KZ679151	366	1101	40.49	4.89	32.96	72.57	−0.269	No	No	extracellular

Gene IDs were derived from NCBI’s gene IDs. Yes: the sequence has a signal peptide or transmembrane domain. No: the sequence does not have a signal peptide or transmembrane domain.

**Table 2 jof-11-00714-t002:** Ten conserved motif sequences of the T6 GH18 family proteins.

Motifs	Width (aa)	Motifs Sequence
Motif 1	36	SHVNLTETDBAVSMYTKAGVPANKIVLGIPSYGRSF
Motif 2	21	SVVRFLGNYGLDGIDIDWEYP
Motif 3	48	FNKKTWGWNGCPNLLVGYKICLSKGSPMPAPVPNAVCGPQKPGTKRP
Motif 4	29	AFKYJDLAAIDQYLDYINLMAYDYAGSWD
Motif 5	15	AGVKVJLSJGGWTYS
Motif 6	14	ITHVIYAFANITED
Motif 7	18	LNPCPLNACCNIWGQCGT
Motif 8	29	EEEAEBYLLLLREVRKALPADAALSIAVP
Motif 9	37	YNDTQWVAYMSDKTKASRIDKYPALNFLGIVDWAIDL
Motif 10	32	FCTPAKSSTGAPGTAPPGENGCISNCGTSIVT

Width (aa): number of amino acids included in the motif. The results were obtained by MEME.

**Table 3 jof-11-00714-t003:** Secondary structure of the T6 GH18 family proteins.

Proteins	Alpha Helix(%)	Extended Strand (%)	Beta Turn (%)	Random Coil(%)	Distribution of Secondary Structure Elements
GH18-1	27.43	15.71	0.00	56.86	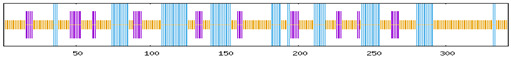
GH18-2	20.30	10.40	0.00	69.29	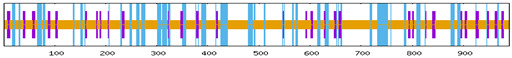
GH18-3	31.46	17.13	0.00	51.40	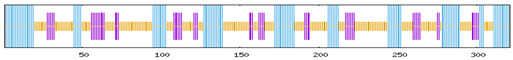
GH18-4	15.80	14.86	0.00	69.34	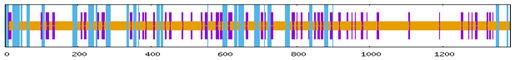
GH18-5	26.61	14.33	0.00	59.06	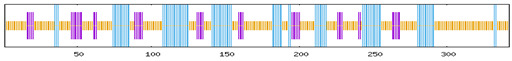
GH18-6	27.70	16.03	0.00	56.27	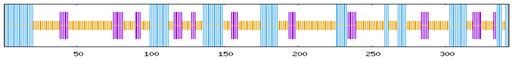
GH18-7	28.97	15.37	0.00	55.67	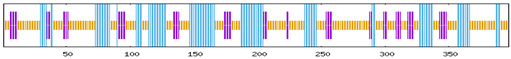
GH18-8	27.18	14.87	0.00	57.95	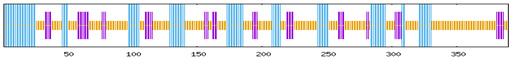
GH18-9	32.72	11.69	0.00	55.59	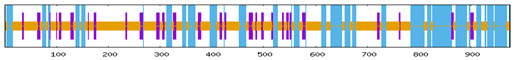
GH18-10	14.50	13.91	0.00	71.59	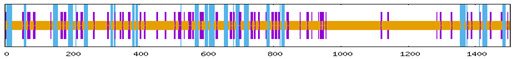
GH18-11	25.56	13.65	0.00	60.79	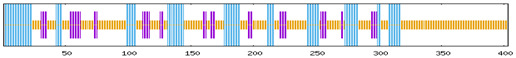
GH18-12	36.64	16.28	0.00	47.07	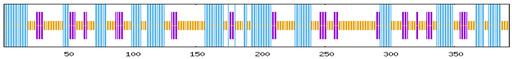
GH18-13	29.69	17.93	0.00	52.38	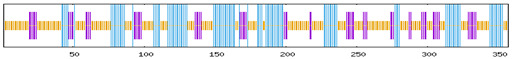
GH18-14	26.79	15.05	0.00	58.16	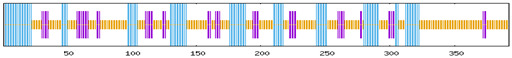
GH18-15	23.66	16.10	0.00	60.24	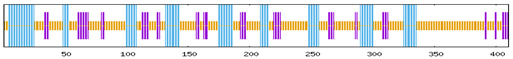
GH18-16	32.55	14.86	0.00	52.59	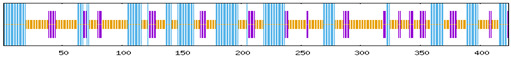
GH18-17	23.58	13.62	0.00	62.80	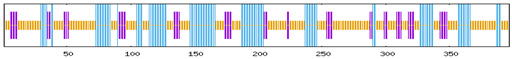
GH18-18	28.42	17.49	0.00	54.10	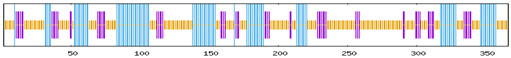

## Data Availability

The original contributions presented in the study are included in the article; further inquiries can be directed to the corresponding authors/first author.

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
