# Peer review of "Genome-Wide Identification and Analysis of Chitinase GH18 Gene Family in Trichoderma longibrachiatum T6 Strain: Insights into Biocontrol of Heterodera avenae"

_jof, 2025, doi:10.3390/jof11100714_

Round 1
Reviewer 1 Report
The article provides a comprehensive analysis of the organization, structure, and expression of the GH18 family genes in Trichoderma longibrachiatum T6, with a focus on their role in controlling the population of Heterodera avenae. This work is highly relevant as it aims to find environmentally friendly methods for controlling nematodes and demonstrates a wide range of bioinformatics approaches. However, there are several questions and comments regarding this article.
1.
The text of the article contains numerous grammatical and stylistic errors (for example, “have been act as one of the key mechanisms”, “in controlling of Heterodera avenae”) . This makes the material difficult to understand and reduces the scientific level of the article. I recommend that a professional language editor review the article.
2. The analysis in the work is mainly limited to bioinformatics methods (identification, phylogeny, structure prediction, and qRT-PCR expression). However, there are no functional experiments (e.g., knockout/overexpression of GH18-3 and GH18-16) that would confirm their actual role in biocontrol. This weakens the applied significance of the findings and their importance in science.
3. The Materials and Methods section contains too many technical details about the use of software (TBtools, MEME, SignalP, etc.). For journal format, it is recommended to reduce the description of tools and move them to Supplementary, leaving the basic analysis parameters sufficient for reproduction.
4. In the Discussion section, the authors repeat the results, but they do not compare their data with published studies on other Trichoderma species. For example, when discussing the similarities between GH18-3 and Chit33 and GH18-16 and Chit42, it would have been helpful to explore the differences in expression and activity between the species, which would have strengthened the role of these genes in the study.
5. The conclusion states that the work "establishes a foundation for the development of new nematicidal genes," but no specific proposals are presented for translating this data into applied technologies (such as the creation of transgenic plants, the development of biologics, etc.). The "Conclusions" section should be strengthened with practical perspectives.
Author Response
Response to Reviewer #1
Dear Reviewer
We sincerely appreciate your efforts in handling the review of our manuscript. We have submitted a revised version of our manuscript titled "Genome-Wide Identification and Analysis of Chitinase GH18 Gene Family in Trichoderma longibrachiatum T6 Strain: Insights into Biocontrol of Heterodera avenae" .We are grateful for your constructive comments on our study and have accepted and implemented the suggested revisions in the revised manuscript. All changes have been highlighted in red. Below is a detailed list of our responses to all comments and critiques, as well as the changes we have made. If you have any further questions, please do not hesitate to let us know. We extend our heartfelt thanks for your positive consideration of our manuscript.
With best wishes,
Yours sincerely,
Cizhong Duan
Major comment:
The article provides a comprehensive analysis of the organization, structure, and expression of the GH18 family genes in Trichoderma longibrachiatum T6, with a focus on their role in controlling the population of Heterodera avenae. This work is highly relevant as it aims to find environmentally friendly methods for controlling nematodes and demonstrates a wide range of bioinformatics approaches. However, there are several questions and comments regarding this article.
Response:
Thank you for your valuable suggestions. We would like to thank you for your efforts in scrutinizing the thesis and offering valuable suggestions, which help to improve the quality of this manuscript. According to your valuable comments, we have made a major revision and incorporated your comments in the preparation of the revised version of our manuscript.
Detailed comments
Comment 1:
1) The text of the article contains numerous grammatical and stylistic errors (for example, “have been act as one of the key mechanisms”, “in controlling of Heterodera avenae”). This makes the material difficult to understand and reduces the scientific level of the article. I recommend that a professional language editor review the article.
Response:
We are deeply grateful for your contributions to the improvement of this paper. Accordingly, we have made careful and detailed revisions based on your guidance. The relevant changes are as follows:
L11-12: “Cereal cyst nematode (Heterodera avenae) causes significant economic losses to global wheat production.” was changed to “Heterodera avenae (Tylenchida: Heteroderidae) causes significant economic losses in the production of wheat, barley, and other cereal crops worldwide.”
L12-16: “The extracellular enzymes particularly glycoside hydrolase 18 (GH18) family chitinase that secreted from the Trichoderma species, have been act as one of the key mechanisms for nematodes controlling. However, information related to Trichoderma longibrachiatum T6 (T6) GH18 family genes in controlling of Heterodera avenae was unexplored by genome-wide analysis.” was changed to “Extracellular enzymes, particularly those from the glycoside hydrolase 18 (GH18) family, such as chitinases secreted by Trichoderma (Moniliales: Moniliaceae) spp., play a crucial role in nematode control. However, the genome-wide analysis of Trichoderma longibrachiatum T6 (T6) GH18 family genes in the context of controlling H. avenae remains unexplored.”
L16-18: “This study identified and thoroughly analyzed the gene structure, evolutionary development, protein characteristics, and gene expression profiles following parasitism with H. avenae cysts in 18 GH18 family members distributed across 13 chromosomes.” was changed to “Through phylogenetic analysis and bioinformatics tools, we identified and conducted a detailed analysis of 18 GH18 genes distributed across 13 chromosomes.”
L20-26: “Our results found that 18 GH18 members in T6 were clustered into three major groups (A, B, C) that comprising seven subgroups, each exhibiting highly conserved catalytic domains, motifs, and gene structures, as well as the cis-acting element showed extensive responses to hormones, stress-related and light signals; significantly enriched in the chitin catabolic process, extracellular region, and chitinase activity (GO functional enrichment); and involved in amino sugar and nucleotide sugar metabolism( KEGG pathway enrichment), whereas 13 members formed an interaction network, and enhanced in chitin degradation efficiency through synergistic effects.” was changed to “Our results indicate that 18 GH18 members in T6 were clustered into three major groups (A, B, C), which comprise seven subgroups. Each subgroup exhibits highly conserved catalytic domains, motifs, and gene structures, while the cis-acting elements show extensive responses to hormones, stress-related signals, and light. These members are significantly enriched in the chitin catabolic process, extracellular region, and chitinase activity (GO functional enrichment), and they are involved in amino sugar and nucleotide sugar metabolism (KEGG pathway enrichment).”
L28-31: “Interestingly, 18 members of GH18 family genes was expressed after the T6 parasitism on H. avenae cysts, especially the GH18-3 (Group B) and GH18-16 (Group A) were significantly upregulated, and which were upregulated averagely by 3.21-folds and 3.10-folds from 12 to 96 hours after parasitism in comparison to control, respectively.” was changed to “Interestingly, 18 members of the GH18 family genes were expressed after T6 parasitism on H. avenae cysts. Notably, GH18-3 (Group B) and GH18-16 (Group A) were significantly upregulated, with average increases of 3.21-fold and 3.10-fold, respectively, from 12 to 96 h after parasitism when compared to the control group.”
L34-36: “Our results indicate that the T6 GH18 family genes play a critical role in cyst parasitism, and even establish a foundation for exploring and identifying some novel nematicidal genes from Trichoderma.” was changed to “Our results indicate that the T6 GH18 family genes play a critical role in cyst parasitism and establish a foundation for the exploration and identification of novel nematicidal genes from Trichoderma.”
L41-47: “Plant-parasitic nematodes (PPNs) are significant economic pathogens that cause diseases in plants, impeding their growth and production. It is estimated that they leading to economic losses up to US $ 215.77 billion per year in global agricultural production [1]. Heterodera avenae is one of the most economical and significant plant-parasitic nematodes [2] that causing substantial economic losses in major wheat-producing regions worldwide [3].” was changed to “Plant-parasitic nematodes (PPNs) are among the most significant soil-borne pathogens that cause diseases in plants, thereby impeding their growth and production [1]. It is estimated that global agricultural production incurs losses of up to $215.77 billion annually due to PPNs [2]. Heterodera avenae is one of the PPNs that poses serious threat to wheat production [3] and causes substantial economic losses in major wheat-producing regions worldwide (over 40 countries and regions including North Africa, West Asia, India, China, Australia, and Europe) [4].”
L63-65: “Our previous research demonstrated that T. longibrachiatum T6 exhibits significant parasitism and lethality against H. avenae cysts, which is associated with an increase in chitinase activity, contributing to its parasitic and lethal effects [11].” was changed to “Our previous studies have demonstrated that T6 exhibits strong parasitic and lethal effects on H. avenae by the enhanced extracellular chitinase activity [19–21].”
L441-445: “Gene structure analysis reveals that Group A genes contain more introns than those in Groups B and C. This suggests their potential to enhance the evolution of biological functions through recombination and increased complexity during the evolutionary process [50].” was changed to “Gene structure analysis reveals that the genes in Group A contain a greater number of introns compared to those in Groups B and C. This finding suggests that Group A genes may enhance the evolution of biological functions through mechanisms such as recombination and increased complexity during the evolutionary process [61].”
L455-457: “The specific distribution of developmental and cellular process regulatory elements in the promoter regions of GH18 family genes reveals the potential functions of these genes in the growth and development of T6.” was changed to “The specific distribution of developmental and cellular process regulatory elements within the promoter regions of GH18 family genes suggests their potential roles in the growth and development of T6.”
L491-494: “These findings collectively suggest that GH18-3 and GH18-16 may possess the capability to degrade the cell walls of pathogenic fungi and the cyst walls of nematodes, thereby endowing it with potential dual functions of antimicrobial and nematicidal activities.” was changed to “These findings collectively suggest that GH18-3 and GH18-16 may possess the capability to degrade the cell walls of pathogenic fungi and the cyst walls of nematodes, thereby endowing them with potential dual functions of antimicrobial and nematicidal activities.”
Comment 2:
2) The analysis in the work is mainly limited to bioinformatics methods (identification, phylogeny, structure prediction, and qRT-PCR expression). However, there are no functional experiments (e.g., knockout/overexpression of GH18-3 and GH18-16) that would confirm their actual role in biocontrol. This weakens the applied significance of the findings and their importance in science.
Response:
We are deeply grateful for your meticulous review of our work and the valuable feedback you have provided. We fully agree with your observation that the current study is primarily confined to bioinformatics analyses, including gene identification, phylogenetic analysis, structural prediction, and qRT-PCR expression analysis, and lacks functional experiments, such as the knockout or overexpression of GH18-3 and GH18-16, to validate their actual roles in biocontrol. Indeed, we recognize the importance of functional verification in elucidating the roles of GH18 family members in antagonism against Heterodera avenae. Currently, we have initiated the construction of knockout mutants and overexpression strains for GH18-3 and GH18-16, and we plan to conduct follow-up experiments in the near future. We will focus on the results of the aforementioned functional verification as the basis for our subsequent research and present them in a separate research paper, aiming to further supplement and improve the shortcomings of our current work. Once again, we thank you for your attention to our research and your constructive comments, which have significantly guided us in enhancing the depth and breadth of our work.
Comment 3:
3) The Materials and Methods section contains too many technical details about the use of software (TBtools, MEME, SignalP, etc.). For journal format, it is recommended to reduce the description of tools and move them to Supplementary, leaving the basic analysis parameters sufficient for reproduction.
Response:
Thank you for your careful review and constructive suggestions regarding our manuscript. Based on your suggestion, we have added Table S1 in the 'Materials and Methods' section, which provides a detailed description of the bioinformatics tool websites. The related revised file of Table S1 is as follow:
Table S1 Bioinformatics analysis websites in this study.
Software names |
Access time |
Online website |
NCBI database |
24 April 2025 |
http://www.ncbi.nlm.nih.gov/ |
Pfam database |
24 April 2025 |
http://pfam.xfam.org/ |
SMART database |
24 April 2025 |
http://smart.embl-heidelberg.de/ |
ExPASy |
24 April 2025 |
https://web.expasy.org/compute pi/ |
WoLF PSORT |
24 April 2025 |
https://wolfpsort.hgc.jp/ |
SignalP-5.0 |
25 April 2025 |
https://services.healthtech.dtu.dk/service.php?SignalP-5.0 |
Deep TMHMM tool |
25 April 2025 |
https://dtu.biolib.com/DeepTMHMM |
NetPhos 3.1 |
25 April 2025 |
http://www.cbs.dtu.dk/services/NetPhos/ |
PROSITE database |
25 April 2025 |
https://prosite.expasy.org/ |
MEME Suite |
27 April 2025 |
http://memesuite.org/tools/meme |
Phytozome databases |
27 April 2025 |
https://phytozome.jgi.doe.gov/ |
iTOL server |
27 April 2025 |
https://itol.embl.de/ |
PlantCARE database |
28 April 2025 |
http://bioinformatics.psb.ugent.be/webtools/plantcare/html/ |
eggNOG-mapper |
28 April 2025 |
http://eggnog5.embl.de/ |
WEGO 2.0 |
28 April 2025 |
https://wego.genomics.cn/ |
KEGG database |
29 April 2025 |
https://www.kegg.jp/kegg/pathway.html |
NPS@SOPMA |
29 April 2025 |
https://npsa.lyon.inserm.fr/cgi-bin/npsa_automat.pl?page=/NPSA/npsa_sopma.html |
UniProt |
29 April 2025 |
https://www.uniprot.org/ |
STRING |
30 April 2025 |
https://cn.string-db.org/ |
Comment 4:
4) In the Discussion section, the authors repeat the results, but they do not compare their data with published studies on other Trichoderma species. For example, when discussing the similarities between GH18-3 and Chit33 and GH18-16 and Chit42, it would have been helpful to explore the differences in expression and activity between the species, which would have strengthened the role of these genes in the study.
Response:
Thank you most sincerely for contributing to the improvement of this manuscript. Through your guidance, we have recognized the necessity of comparing our data with studies on the GH18 family in other published species(Trichoderma spp.). The related statements were as follows:
L396: “Subcellular localization predictions indicated that, with the exception of two proteins in the A-IV subgroup predicted to localize to mitochondria, all others were extracellular.” was deleted.
L402: “with five chromosomes each containing two genes, while the remaining eight chromosomal segments housed one gene each.” was deleted.
L477: “Secondary structure prediction revealed that all GH18 chitinases possess conserved alpha helices (14.50%-36.64%), extended strands (10.40%-17.93%), and random coils (47.07%-71.59%). Among these structures, random coils predominate across all isoforms, achieving an evolutionary balance between conformational flexibility and catalytic rigidity. The functional specificity of proteins is determined by their tertiary structure [55]. The tertiary structures of all GH18 members adopt the canonical TIM (βα)8-barrel (also known as TIM barrel) folding pattern. Within each subgroup, the three-dimensional structures are highly conserved; however, notable structural differences are evident across subgroups.” was deleted.
L397-401: “Within the GH18 family of T6, 11 members possess a signal peptide, whereas 15 members in T. reesei exhibit this characteristic as well, indicating that these proteins are likely secreted [52]. The lack of a typical N-terminal signal peptide in the remaining members suggests that they may be secreted through non-classical pathways [54].” was added.
L402-406: “In this study, the GH18 genes demonstrated a conserved evolutionary arrangement across chromosomes. This distribution pattern is consistent with the chromosomal organization observed in the GH18 families across most Trichoderma spp., likely underpinning their functional diversification and transcriptional regulation [55].” was added.
L424-426: “Similarly, the majority of C-I and C-II in the fungal GH18 family demonstrate significant interspecies sequence variability, which is driven by positive selection and diversification evolution [48].” was added.
L465-468: “This is similar to the GO functional annotation of the GH18 family T6-Echi18-5 protein in T6, as previously reported by Shen et al., which plays a significant role in the degradation of H. avenae eggs by T6 [30].” was added.
L486-489: “In the presence of pathogenic fungi (R. solani), which contain a high proportion of chitin, the GH18 family group A genes AO-190 and AO-801 of the nematode-trapping fungus A. oligospora exhibit enhanced expression. This observation indicates their potential involvement in biological control and mycoparasitism processes [31].” was added.
L536-543: “This discovery closely aligns with the transcriptional expression profiles of the GH18 family genes chit18-2 (Group A), chi18-4 (Group A), chi18-13 (Group A), chi18-3 (Group B), and chi18-10 (Group C) in T. atroviride, whose expression is influenced by varying growth conditions and carbon sources [57]. This suggests that the functions of GH18 family chitinases in Trichoderma spp. are diverse; they are not mutually substitutable but rather fulfill specific roles within the organism. Similarly, the GH18 family genes in A. oligospora exhibit differential expression at various growth stages and perform multiple functions throughout growth, differentiation and infection [41].” was added.
Comment 5:
5) The conclusion states that the work "establishes a foundation for the development of new nematicidal genes," but no specific proposals are presented for translating this data into applied technologies (such as the creation of transgenic plants, the development of biologics, etc.). The "Conclusions" section should be strengthened with practical perspectives.
Response:
Thank you for your comments and providing good suggestions for my manuscript. We have revised the "Conclusion" section based on your suggestions, strengthening the perspective of practical application technologies. The relevant changes are as follows:
Bioinformatic analysis revealed the evolutionary relationships and conserved characteristics of the GH18 gene family in T6, while expression quantification analysis elucidated the regulatory mechanisms of these genes during the parasitism of H. avenae cysts. Crucially, GH18-3 and GH18-16 were identified as core genes interacting with nematodes, providing essential targets for in-depth exploration of the molecular mechanisms by which Trichoderma spp. chitinases antagonize H. avenae parasitism. The findings of this study not only deepen the fundamental theoretical research on chitinase-based biopesticides but also provide an external impetus for the genetic engineering modification of strains, thereby promoting the advancement of Trichoderma field control technologies. Furthermore, this research offers new insights into the development of environmentally friendly strategies for controlling PPNs.

Reviewer 2 Report
Manuscript is a good pice of science however, it need many amendments to be published.
It is necessary more references in introduction.
Second paragraph is too large and has no flow. Divide it into three ones.
You indicate a gap of knowledge but there is no a clear justification. Add one to justify your manuscript.
Along manuscript you begin several statements with a scientific name, but you must indicate “The species…” instead of that.
Also, at first time indicate a scientific name add vernacular name according to Encyclopedia of life and after the scientific name add (Order: Family).
Genome-Wide Identification and Analysis of Chitinase GH18 Gene Family in Trichoderma longibrachiatum T6 Strain: Insights into Biocontrol of Heterodera avenae
L.12: State the reason why "Heterodera avenae causes these million-dollar losses" (which affects the crop).
L.16: State or mention the websites or applications used in this work.
L.36-37: Remove the terms "Trichoderma longibrachiatum; Heterodera avenae" from the keywords because they are part of the title.
L.40: Explain the term “economic pathogens.”
L.40-44: The term “economic” is repeated frequently in this paragraph, making it redundant.
L.43: The term “is one of the most economical and significant plant parasites” sounds confusing in the statement.
L.43: Explain what you mean by “most economical plant parasites.” They mean they cause economic losses or are cheap to purchase.
L.44: Indicate the wheat-producing regions worldwide.
L.48: Indicate what you mean by “eco-efficient.”
L.50: Include a line of text introducing why Trichoderma spp. is important in biocontrol.
L.50-81: This paragraph is too long; this makes reading tiring. Divide this paragraph into two or three paragraphs; this will make reading more comfortable.
L.58: Indicate what you mean by "significant lethality," since the word "significant" is ambiguous.
L.521-526: The statement in these lines is intended more as a summary of the work done, so it is not covered in this section.
Author Response
Response to Reviewer #2
Dear Reviewer
We sincerely appreciate your efforts in handling the review of our manuscript. We have submitted a revised version of our manuscript titled "Genome-Wide Identification and Analysis of Chitinase GH18 Gene Family in Trichoderma longibrachiatum T6 Strain: Insights into Biocontrol of Heterodera avenae" .We are grateful for your constructive comments on our study and have accepted and implemented the suggested revisions in the revised manuscript. All changes have been highlighted in red. Below is a detailed list of our responses to all comments and critiques, as well as the changes we have made. If you have any further questions, please do not hesitate to let us know. We extend our heartfelt thanks for your positive consideration of our manuscript.
With best wishes,
Yours sincerely,
Cizhong Duan
Major comment:
Manuscript is a good pice of science however, it need many amendments to be published. It is necessary more references in introduction. Second paragraph is too large and has no flow. Divide it into three ones. You indicate a gap of knowledge but there is no a clear justification. Add one to justify your manuscript. Along manuscript you begin several statements with a scientific name, but you must indicate “The species…” instead of that. Also, at first time indicate a scientific name add vernacular name according to Encyclopedia of life and after the scientific name add (Order: Family).
Response:
Thank you for your valuable suggestions. We would like to thank you for your efforts in scrutinizing the thesis and offering valuable suggestions, which help to improve the quality of this manuscript. According to your valuable comments, we have made a major revision and incorporated your comments in the preparation of the revised version of our manuscript. Our response to each of the comments is below the comment in italics. The suggested revisions strengthened the manuscript.
Comment 1:
- It is necessary more references in introduction.
Response:
We are deeply grateful for your contributions to the improvement of this paper. Accordingly, we have made careful and detailed revisions based on your guidance. Additionally, we have enriched the introduction with pertinent literature. The specific changes made are outlined below:
Zhang, S.; Gan, Y.; Ji, W.; Xu, B. Biocontrol potential of a native species of Trichoderma longibrachiatum against Meloidogyne incognita. Appl. Soil Ecol. 2015, 94, 21-29.
Schuster, A.; Schmoll, M. Biology and biotechnology of Trichoderma. Appl. Microbiol. Biotechnol. 2010, 87, 787–799.
Sood, M.; Kapoor, D.; Kumar, V.; Sheteiwy, M.S.; Ramakrishnan, M.; Landi, M.; Araniti, F. Trichoderma: The “secrets” of a multitalented biocontrol agent. Plants 2020, 9, 762.
Hanhong, B. Trichoderma Species as abiotic and biotic stress quenchers in Plants. Res. J. Biotechnol. 2011, 6, 73–79.
Bai, B.; Liu, C.; Zhang, C.; Ai, H.; Xiao, L.; Liu, H.; Chen, J.; Zhang, Y.; Zhang, W.; Chen, W. Trichoderma species from plant and soil: an excellent resource for biosynthesis of terpenoids with versatile bioactivities. J. Adv. Res. 2023, 49, 81–102.
Howell, C.R. Mechanisms Employed by Trichoderma species in the biological control of plant diseases: The history and evolution of current concepts. Plant Dis. 2003, 87, 4–10.
Qualhato, T.F.; Lopes, F.A.; Steindorff, A.S.; Brandão, R.S.; Jesuino, R.S.; Ulhoa, C.J. Mycoparasitism studies of Trichoderma species against three phytopathogenic fungi: Evaluation of antagonism and hydrolytic enzyme production. Biotechnol. Lett. 2013, 35, 1461–1468.
Lopes, F.A.; Steindorff, A.S.; Geraldine, A.M.; Brandão, R.S.; Monteiro, V.N.; Junior, M.L.; Coelho, A.S.; Ulhoa, C.J.; Silva, R.N. Biochemical and metabolic profiles of Trichoderma strains isolated from common bean crops in the Brazilian Cerrado, and potential antagonism against Sclerotinia sclerotiorum. Fungal Biol. 2012, 116, 815–824.
Chacón, M.R.; Rodríguez-Galán, O.; Benítez, T.; Sousa, S.; Rey, M.; Llobell, A.; Delgado-Jarana, J. Microscopic and transcriptome analyses of early colonization of tomato roots by Trichoderma harzianum. Int. Microbiol. 2007, 10, 19–27.
Metcalf, D.A.; Wilson, C.R. The process of antagonism of Sclerotium cepivorum in white rot affected onion roots by Trichoderma koningii. Plant Pathol. 2001, 50, 249–257.
Álvarez, García S; Mayo, Prieto S; Carro, Huerga, G. Volatile organic compound chamber: a novel technology for microbiological volatile interaction assays. J. Fungi 2021, 7, 248.
Zhang, S.; Yantai, Z.; et al. The parasitic and lethal effects of Trichoderma longibrachiatum against Heterodera avenae. Biol. Control 2014, 2, 1–8.
Zhang, S.; Gan, Y.; Liu, J.; Zhou, J.; Xu, B. Optimization of the fermentation media and parameters for the biocontrol potential of Trichoderma longibrachiatum T6 against Nematodes. Front. Microbiol. 2020, 11, 574601.
Zhang, S.; Gan, Y.; Ji, W.; Xu, B.; Hou, B.; Liu, J. Mechanisms and Characterization of Trichoderma longibrachiatum T6 in Suppressing Nematodes (Heterodera avenae) in wheat. Front. Plant Sci. 2017, 8, 1491.
Yang, J.; Yu, Y.; Li, J.; Zhu, W.; Geng, Z.; Jiang, D.; Zeng, W.; Zou, C. Characterization and functional analyses of the chitinase encoding genes in the nematode-trapping fungus Arthrobotrys oligospora. Arch. Microbiol. 2013, 195, 453–462.
Gruber, S.; Kubicek, C.P.; Seidl-Seiboth, V. Differential regulation of orthologous chitinase genes in mycoparasitic Trichoderma species. Appl. Environ. Microbiol. 2011, 77, 7217–7226.
Comment 2:
2) Second paragraph is too large and has no flow. Divide it into three ones.
Response:
We deeply appreciate your contributions to the improvement of this article. Accordingly, we have conducted careful and detailed revisions based on your guidance. Furthermore, we have divided the second paragraph of the introduction section into three distinct paragraphs to enhance clarity and coherence. The specific modifications are outlined below:
Trichoderma spp. are beneficial fungi that recognized for their broad-spectrum activity, safety, adaptability, and diverse mechanisms of action [7–11]. They demonstrate considerable antagonistic activity against pathogens [12,13] and can efficiently compete for nutritional resources [14,15], establishing them as one of the most effective microbial resources for biological control [16]. One of the main mechanisms for Trichoderma spp. against nematodes was mainly relies on various degrading enzymes (chitinase, glucanase, and xylanase, etc.) production and secretion [17], which are beneficial for hyphae to penetrate the eggshells of nematodes or the cuticles of juveniles and adults to colonize, and finally leading to the degradation of the body wall and the death of the nematodes [5]. Among them, chitinase plays a central role in the process of parasitism and degradation of the nematode body wall [18]. Our previous studies have demonstrated that T6 exhibits strong parasitic and lethal effects on H. avenae by the enhanced extracellular chitinase activity [19–21]. The chitinase-encoding genes of Trichoderma spp. predominantly belong to the GH18 multigene family [22], which is characterized by a GH18 catalytic domain and several auxiliary domains [23]. Members of this family have been confirmed as key antagonistic factors in the species including ThEn-42 from T. atroviride [24], chit33 [25], chit36 [26], Chit37 [27], and Chit42 [28] from T. harzianum. Specifically, ThEn-42 is involved in the biocontrol of Penicillium digitatum (Eurotiales :Aspergillaceae), chit33 targets Rhizoctonia solani (Thanatephorus: Ceratobasidiaceae), chit36 is effective against Botrytis cinerea (Helotiales: Sclerotiniaceae), Chit37 is directed against Fusarium oxysporum (Hypocreales: Nectriaceae), and Chit42 combats Sclerotinia sclerotiorum (Helotiales: Sclerotiniaceae).
Previous studies have primarily concentrated on the species and functions of GH18 family genes in Trichoderma spp., as well as their roles in combating plant pathogens and controlling fungal diseases in plants. However, there is a paucity of information regarding nematodes, particularly PPNs. Studies have demonstrated that the GH18 family genes chit42 and chit33 of T. harzianum are significantly upregulated during the parasitism of Caenorhabditis elegans (Rhabditida: Rhabditidae) eggs, indicating that their chitinases may facilitate the degradation of nematode eggshell [29]; the GH18 family chitinase gene T6-Echi18-5 from T6 was markedly upregulated during the infection of H. avenae cysts, and that the recombinant protein derived from this gene effectively degrades the eggshell and inhibits egg hatching [30].
Current research predominantly focuses on the functional validation of individual genes. In contrast, Yang et al. discovered that genes in the GH18 family of Arthrobotrys oligospora (Orbiliales: Orbiliaceae) exhibit differential expression across various growth stages and demonstrate diverse regulatory functions during growth, differentiation, and pathogen infection processes[31]. Given the complexity of the functional regulation of this gene family, systematic functional identification and analysis of regulatory mechanisms are particularly crucial. However, there is still a lack of systematic research on the T6 GH18 gene family, particularly regarding which members are involved in the transcriptional regulation of nematocidal mechanisms during the parasitism of the H. avenae cysts, which urgently requires in-depth investigation.
Comment 3:
3) You indicate a gap of knowledge but there is no a clear justification. Add one to justify your manuscript.
Response:
Thank you for reviewing our manuscript and for the constructive comments, which greatly helped us to improve the manuscript. We have supplemented the origin according to your suggestions to refine the introduction section of this manuscript. The relevant changes are as follows:
L74-77: “Previous studies have primarily concentrated on the species and functions of GH18 family genes in Trichoderma spp., as well as their roles in combating plant pathogens and controlling fungal diseases in plants. However, there is a paucity of information regarding nematodes, particularly PPNs.” was added.
L84-90: “Current research predominantly focuses on the functional validation of individual genes. In contrast, Yang et al. discovered that genes in the GH18 family of Arthrobotrys oligospora (Orbiliales: Orbiliaceae) exhibit differential expression across various growth stages and demonstrate diverse regulatory functions during growth, differentiation, and pathogen infection processes[31]. Given the complexity of the functional regulation of this gene family, systematic functional identification and analysis of regulatory mechanisms are particularly crucial.” was added.
Comment 4:
4) Along manuscript you begin several statements with a scientific name, but you must indicate “The species…” instead of that.
Response:
Thank you for your kind letter and your careful work regarding our manuscript. In response to your suggestion, we have replaced the specific species names in the manuscript with the phrase "The species...". Furthermore, we have simplified the reference to the Trichoderma longibrachiatum T6 strain to simply "T6" throughout the manuscript. Here are the point-by-point responses for the reviewers’ comments.
L67-69: “Members of this family have been confirmed as key antagonistic factors in Trichoderma spp. .” was changed to “Members of this family have been confirmed as key antagonistic factors in the species including ThEn-42 from T. atroviride [24], chit33 [25], chit36 [26], Chit37 [27], and Chit42 [28] from T. harzianum.”
L415-417: “Phylogenetic analysis clustered the 18 T6 GH18 chitinases into three groups (A, B, and C), which were further subdivided into seven subgroups. This classification is consistent with previous studies on Trichoderma spp. [12,41].” was changed to “Phylogenetic analysis clustered the 18 T6 GH18 chitinases into three groups (A, B, and C), which were further subdivided into seven subgroups. This classification is consistent with previous studies on the species [22,52].”
Comment 5:
5) Also, at first time indicate a scientific name add vernacular name according to Encyclopedia of life and after the scientific name add (Order: Family).
Response:
Thank you for your comments and providing good suggestions for my manuscript. Therefore, we have revised the scientific names by adding (Order: Family) after them, in accordance with your instructions. The relevant changes are as follows:
Heterodera avenae (Tylenchida: Heteroderidae) ï¼›Trichoderma (Moniliales: Moniliaceae)ï¼›Penicillium digitatum (Eurotiales :Aspergillaceae)ï¼›Rhizoctonia solani (Thanatephorus: Ceratobasidiaceae)ï¼›Botrytis cinerea (Helotiales: Sclerotiniaceae)ï¼›Fusarium oxysporum (Hypocreales: Nectriaceae)ï¼›Sclerotinia sclerotiorum (Helotiales: Sclerotiniaceae)ï¼›Caenorhabditis elegans (Rhabditida: Rhabditidae)ï¼›Arthrobotrys oligospora (Orbiliales: Orbiliaceae)ï¼›Aspergillus nidulans (Eurotiales: Aspergillaceae)
Detailed comments
Comment 1:
1) L.12: State the reason why "Heterodera avenae causes these million-dollar losses" (which affects the crop).
Response:
We are deeply grateful for your contributions to the improvement of this paper. My point-by-point explanation of the comments is as follows:
Heterodera avenae primarily infects the roots of cereal crops, notably wheat and barley. The root systems of infected plants exhibit a reduction in length and an increase in thickness, yet do not display symptoms of root knots, resulting in an impaired ability to absorb water and nutrients from the soil. The symptoms observed on the leaves of infected plants share many characteristics with those of root diseases. The above-ground portions of the plants demonstrate weakened growth, stunting, yellowing, reduced tillering, and sparse plant distribution across the field, resembling symptoms associated with certain non-infectious diseases. ( References: Smiley R W, Yan G. Cereal cyst nematodes biology and management in pacific northwest wheat, barley, and oat crops[J]. A Pacific Northwest Extension Publication, 2010, pnw620.)
Comment 2:
2) L.16: State or mention the websites or applications used in this work.
Response:
We are deeply grateful for your meticulous review of our work and the valuable feedback you have provided. Based on your suggestion, we have added Table S1 in the 'Materials and Methods' section, which provides a detailed description of the bioinformatics tool websites. The related revised file of Table S1 is as follow:
Table S1 Bioinformatics analysis websites in this study.
Software names |
Access time |
Online website |
NCBI database |
24 April 2025 |
http://www.ncbi.nlm.nih.gov/ |
Pfam database |
24 April 2025 |
http://pfam.xfam.org/ |
SMART database |
24 April 2025 |
http://smart.embl-heidelberg.de/ |
ExPASy |
24 April 2025 |
https://web.expasy.org/compute pi/ |
WoLF PSORT |
24 April 2025 |
https://wolfpsort.hgc.jp/ |
SignalP-5.0 |
25 April 2025 |
https://services.healthtech.dtu.dk/service.php?SignalP-5.0 |
Deep TMHMM tool |
25 April 2025 |
https://dtu.biolib.com/DeepTMHMM |
NetPhos 3.1 |
25 April 2025 |
http://www.cbs.dtu.dk/services/NetPhos/ |
PROSITE database |
25 April 2025 |
https://prosite.expasy.org/ |
MEME Suite |
27 April 2025 |
http://memesuite.org/tools/meme |
Phytozome databases |
27 April 2025 |
https://phytozome.jgi.doe.gov/ |
iTOL server |
27 April 2025 |
https://itol.embl.de/ |
PlantCARE database |
28 April 2025 |
http://bioinformatics.psb.ugent.be/webtools/plantcare/html/ |
eggNOG-mapper |
28 April 2025 |
http://eggnog5.embl.de/ |
WEGO 2.0 |
28 April 2025 |
https://wego.genomics.cn/ |
KEGG database |
29 April 2025 |
https://www.kegg.jp/kegg/pathway.html |
NPS@SOPMA |
29 April 2025 |
https://npsa.lyon.inserm.fr/cgi-bin/npsa_automat.pl?page=/NPSA/npsa_sopma.html |
UniProt |
29 April 2025 |
https://www.uniprot.org/ |
STRING |
30 April 2025 |
https://cn.string-db.org/ |
Comment 3:
3) Remove the terms "Trichoderma longibrachiatum; Heterodera avenae" from the keywords because they are part of the title.
Response:
Thank you for your careful review and constructive suggestions regarding our manuscript. In accordance with your suggestion, we have removed "Trichoderma longibrachiatum; Heterodera avenae" from the keywords.
Comment 4:
4) Explain the term “economic pathogens.”
Response:
Thank you most sincerely for contributing to the improvement of this manuscript. I sincerely apologize for any confusion caused by my previous expressions. The related changes were as follows:
L41-42: “Plant-parasitic nematodes (PPNs) are significant economic pathogens that cause diseases in plants, impeding their growth and production.” was changed to “Plant-parasitic nematodes (PPNs) are among the most significant soil-borne pathogens that cause diseases in plants, thereby impeding their growth and production [1].”
Comment 5:
5) L.40-44: The term “economic” is repeated frequently in this paragraph, making it redundant.
Response:
Thank you very much for pointing this out. According to your kindly suggestions, we have revised the frequent occurrence of the term "economic" in this passage. The related changes were as follows:
L41-47: “Plant-parasitic nematodes (PPNs) are significant economic pathogens that cause diseases in plants, impeding their growth and production. It is estimated that they leading to economic losses up to US $ 215.77 billion per year in global agricultural production [1]. Heterodera avenae is one of the most economical and significant plant-parasitic nematodes [2] that causing substantial economic losses in major wheat-producing regions worldwide [3].” was changed to “Plant-parasitic nematodes (PPNs) are among the most significant soil-borne pathogens that cause diseases in plants, thereby impeding their growth and production [1]. It is estimated that global agricultural production incurs losses of up to $215.77 billion annually due to PPNs [2]. Heterodera avenae is one of the PPNs that poses serious threat to wheat production [3] and causes substantial economic losses in major wheat-producing regions worldwide (over 40 countries and regions including North Africa, West Asia, India, China, Australia, and Europe) [4].”
Comment 6:
6) L.43: The term “is one of the most economical and significant plant parasites” sounds confusing in the statement.
Response:
Thank you for your comments and providing good suggestions for my manuscript. We have removed this inappropriate statement.
Comment 7:
7) L.43: Explain what you mean by “most economical plant parasites.” They mean they cause economic losses or are cheap to purchase.
Response:
Thank you most sincerely for contributing to the improvement of this manuscript. My incorrect expression caused confusion in your understanding, and the relevant changes are as follows:
L44-47: “Heterodera avenae is one of the most economical and significant plant-parasitic nematodes [2] that causing substantial economic losses in major wheat-producing regions worldwide [3].” was changed to “Heterodera avenae is one of the PPNs that poses serious threat to wheat production [3] and causes substantial economic losses in major wheat-producing regions worldwide (over 40 countries and regions including North Africa, West Asia, India, China, Australia, and Europe) [4].”
Comment 8:
8) L.44: Indicate the wheat-producing regions worldwide.
Response:
Thank you for your kind letter and your careful work regarding our manuscript. The relevant changes are as follows:
L44-47: “Heterodera avenae is one of the most economical and significant plant-parasitic nematodes [2] that causing substantial economic losses in major wheat-producing regions worldwide [3].” was changed to “Heterodera avenae is one of the PPNs that poses serious threat to wheat production [3] and causes substantial economic losses in major wheat-producing regions worldwide (over 40 countries and regions including North Africa, West Asia, India, China, Australia, and Europe) [4].”
Comment 9:
9) L.48: Indicate what you mean by “eco-efficient.”
Response:
Thank you for your kind letter and your careful work regarding our manuscript. I apologize for any confusion caused by my inappropriate expression. The relevant changes are as follows:
L50-52: “Thus, the development of eco-efficient biological control agents has become a core focus in the management of H. avenae.” was changed to “Thus, the development of efficient and environmentally biological control agents has become the key focus in management of H. avenae.”
Comment 10:
10) L.50: Include a line of text introducing why Trichoderma spp. is important in biocontrol.
Response:
We sincerely appreciate your thorough review of our manuscript and your constructive suggestions. We have revised the manuscript in accordance with your comments and have incorporated the necessary content. The relevant changes are as follows:
L53-57: “Trichoderma spp. are beneficial fungi that recognized for their broad-spectrum activity, safety, adaptability, and diverse mechanisms of action [7–11]. They demonstrate considerable antagonistic activity against pathogens [12,13] and can efficiently compete for nutritional resources [14,15], establishing them as one of the most effective microbial resources for biological control [16].” was added.
Comment 11:
11) L.50-81: This paragraph is too long; this makes reading tiring. Divide this paragraph into two or three paragraphs; this will make reading more comfortable.
Response:
We deeply appreciate your contributions to the improvement of this article. Accordingly, we have conducted careful and detailed revisions based on your guidance. Furthermore, we have divided the second paragraph of the introduction section into three distinct paragraphs to enhance clarity and coherence. The specific modifications are outlined below:
Trichoderma spp. are beneficial fungi that recognized for their broad-spectrum activity, safety, adaptability, and diverse mechanisms of action [7–11]. They demonstrate considerable antagonistic activity against pathogens [12,13] and can efficiently compete for nutritional resources [14,15], establishing them as one of the most effective microbial resources for biological control [16]. One of the main mechanisms for Trichoderma spp. against nematodes was mainly relies on various degrading enzymes (chitinase, glucanase, and xylanase, etc.) production and secretion [17], which are beneficial for hyphae to penetrate the eggshells of nematodes or the cuticles of juveniles and adults to colonize, and finally leading to the degradation of the body wall and the death of the nematodes [5]. Among them, chitinase plays a central role in the process of parasitism and degradation of the nematode body wall [18]. Our previous studies have demonstrated that T6 exhibits strong parasitic and lethal effects on H. avenae by the enhanced extracellular chitinase activity [19–21]. The chitinase-encoding genes of Trichoderma spp. predominantly belong to the GH18 multigene family [22], which is characterized by a GH18 catalytic domain and several auxiliary domains [23]. Members of this family have been confirmed as key antagonistic factors in the species including ThEn-42 from T. atroviride [24], chit33 [25], chit36 [26], Chit37 [27], and Chit42 [28] from T. harzianum. Specifically, ThEn-42 is involved in the biocontrol of Penicillium digitatum (Eurotiales :Aspergillaceae), chit33 targets Rhizoctonia solani (Thanatephorus: Ceratobasidiaceae), chit36 is effective against Botrytis cinerea (Helotiales: Sclerotiniaceae), Chit37 is directed against Fusarium oxysporum (Hypocreales: Nectriaceae), and Chit42 combats Sclerotinia sclerotiorum (Helotiales: Sclerotiniaceae).
Previous studies have primarily concentrated on the species and functions of GH18 family genes in Trichoderma spp., as well as their roles in combating plant pathogens and controlling fungal diseases in plants. However, there is a paucity of information regarding nematodes, particularly PPNs. Studies have demonstrated that the GH18 family genes chit42 and chit33 of T. harzianum are significantly upregulated during the parasitism of Caenorhabditis elegans (Rhabditida: Rhabditidae) eggs, indicating that their chitinases may facilitate the degradation of nematode eggshell [29]; the GH18 family chitinase gene T6-Echi18-5 from T6 was markedly upregulated during the infection of H. avenae cysts, and that the recombinant protein derived from this gene effectively degrades the eggshell and inhibits egg hatching [30].
Current research predominantly focuses on the functional validation of individual genes. In contrast, Yang et al. discovered that genes in the GH18 family of Arthrobotrys oligospora (Orbiliales: Orbiliaceae) exhibit differential expression across various growth stages and demonstrate diverse regulatory functions during growth, differentiation, and pathogen infection processes[31]. Given the complexity of the functional regulation of this gene family, systematic functional identification and analysis of regulatory mechanisms are particularly crucial. However, there is still a lack of systematic research on the T6 GH18 gene family, particularly regarding which members are involved in the transcriptional regulation of nematocidal mechanisms during the parasitism of the H. avenae cysts, which urgently requires in-depth investigation.
Comment 12:
12) L.58: Indicate what you mean by "significant lethality," since the word "significant" is ambiguous.
Response:
Thank you for your comments and providing good suggestions for my manuscript. We have made a correction to the term "significant". The specific modifications are outlined below:
L63-65: “Our previous research demonstrated that T. longibrachiatum T6 exhibits significant parasitism and lethality against H. avenae cysts, which is associated with an increase in chitinase activity, contributing to its parasitic and lethal effects [11].” was changed to “Our previous studies have demonstrated that T6 exhibits strong parasitic and lethal effects on H. avenae by the enhanced extracellular chitinase activity [19–21].”
Comment 13:
13) L.521-526: The statement in these lines is intended more as a summary of the work done, so it is not covered in this section.
Response:
Thank you for your comments and providing good suggestions for my manuscript. We have revised the "Conclusion" section based on your suggestions. The relevant changes are as follows:
Bioinformatic analysis revealed the evolutionary relationships and conserved characteristics of the GH18 gene family in T6, while expression quantification analysis elucidated the regulatory mechanisms of these genes during the parasitism of H. avenae cysts. Crucially, GH18-3 and GH18-16 were identified as core genes interacting with nematodes, providing essential targets for in-depth exploration of the molecular mechanisms by which Trichoderma spp. chitinases antagonize H. avenae parasitism. The findings of this study not only deepen the fundamental theoretical research on chitinase-based biopesticides but also provide an external impetus for the genetic engineering modification of strains, thereby promoting the advancement of Trichoderma field control technologies. Furthermore, this research offers new insights into the development of environmentally friendly strategies for controlling PPNs.
Comment 14:
14) The English could be improved to more clearly express the research.
Response:
We are deeply grateful for your contributions to the improvement of this paper. Thank you very much. As per your suggestion, the entire manuscript has been reviewed by Dr. Solomon Boamah from an English-speaking country (State Key Laboratory of Aridland Crop Science, Gansu Agricultural University, Ghana). He has provided us with many valuable comments and revision suggestions, and we have carefully revised our manuscript based on his recommendations. For more details, please refer to the revised manuscript. The relevant changes are as follows:
L11-12: “Cereal cyst nematode (Heterodera avenae) causes significant economic losses to global wheat production.” was changed to “Heterodera avenae (Tylenchida: Heteroderidae) causes significant economic losses in the production of wheat, barley, and other cereal crops worldwide.”
L12-16: “The extracellular enzymes particularly glycoside hydrolase 18 (GH18) family chitinase that secreted from the Trichoderma species, have been act as one of the key mechanisms for nematodes controlling. However, information related to Trichoderma longibrachiatum T6 (T6) GH18 family genes in controlling of Heterodera avenae was unexplored by genome-wide analysis.” was changed to “Extracellular enzymes, particularly those from the glycoside hydrolase 18 (GH18) family, such as chitinases secreted by Trichoderma (Moniliales: Moniliaceae) spp., play a crucial role in nematode control. However, the genome-wide analysis of Trichoderma longibrachiatum T6 (T6) GH18 family genes in the context of controlling H. avenae remains unexplored.”
L16-18: “This study identified and thoroughly analyzed the gene structure, evolutionary development, protein characteristics, and gene expression profiles following parasitism with H. avenae cysts in 18 GH18 family members distributed across 13 chromosomes.” was changed to “Through phylogenetic analysis and bioinformatics tools, we identified and conducted a detailed analysis of 18 GH18 genes distributed across 13 chromosomes.”
L20-26: “Our results found that 18 GH18 members in T6 were clustered into three major groups (A, B, C) that comprising seven subgroups, each exhibiting highly conserved catalytic domains, motifs, and gene structures, as well as the cis-acting element showed extensive responses to hormones, stress-related and light signals; significantly enriched in the chitin catabolic process, extracellular region, and chitinase activity (GO functional enrichment); and involved in amino sugar and nucleotide sugar metabolism( KEGG pathway enrichment), whereas 13 members formed an interaction network, and enhanced in chitin degradation efficiency through synergistic effects.” was changed to “Our results indicate that 18 GH18 members in T6 were clustered into three major groups (A, B, C), which comprise seven subgroups. Each subgroup exhibits highly conserved catalytic domains, motifs, and gene structures, while the cis-acting elements show extensive responses to hormones, stress-related signals, and light. These members are significantly enriched in the chitin catabolic process, extracellular region, and chitinase activity (GO functional enrichment), and they are involved in amino sugar and nucleotide sugar metabolism (KEGG pathway enrichment).”
L28-31: “Interestingly, 18 members of GH18 family genes was expressed after the T6 parasitism on H. avenae cysts, especially the GH18-3 (Group B) and GH18-16 (Group A) were significantly upregulated, and which were upregulated averagely by 3.21-folds and 3.10-folds from 12 to 96 hours after parasitism in comparison to control, respectively.” was changed to “Interestingly, 18 members of the GH18 family genes were expressed after T6 parasitism on H. avenae cysts. Notably, GH18-3 (Group B) and GH18-16 (Group A) were significantly upregulated, with average increases of 3.21-fold and 3.10-fold, respectively, from 12 to 96 h after parasitism when compared to the control group.”
L34-36: “Our results indicate that the T6 GH18 family genes play a critical role in cyst parasitism, and even establish a foundation for exploring and identifying some novel nematicidal genes from Trichoderma.” was changed to “Our results indicate that the T6 GH18 family genes play a critical role in cyst parasitism and establish a foundation for the exploration and identification of novel nematicidal genes from Trichoderma.”
L41-47: “Plant-parasitic nematodes (PPNs) are significant economic pathogens that cause diseases in plants, impeding their growth and production. It is estimated that they leading to economic losses up to US $ 215.77 billion per year in global agricultural production [1]. Heterodera avenae is one of the most economical and significant plant-parasitic nematodes [2] that causing substantial economic losses in major wheat-producing regions worldwide [3].” was changed to “Plant-parasitic nematodes (PPNs) are among the most significant soil-borne pathogens that cause diseases in plants, thereby impeding their growth and production [1]. It is estimated that global agricultural production incurs losses of up to $215.77 billion annually due to PPNs [2]. Heterodera avenae is one of the PPNs that poses serious threat to wheat production [3] and causes substantial economic losses in major wheat-producing regions worldwide (over 40 countries and regions including North Africa, West Asia, India, China, Australia, and Europe) [4].”
L63-65: “Our previous research demonstrated that T. longibrachiatum T6 exhibits significant parasitism and lethality against H. avenae cysts, which is associated with an increase in chitinase activity, contributing to its parasitic and lethal effects [11].” was changed to “Our previous studies have demonstrated that T6 exhibits strong parasitic and lethal effects on H. avenae by the enhanced extracellular chitinase activity [19–21].”
L441-445: “Gene structure analysis reveals that Group A genes contain more introns than those in Groups B and C. This suggests their potential to enhance the evolution of biological functions through recombination and increased complexity during the evolutionary process [50].” was changed to “Gene structure analysis reveals that the genes in Group A contain a greater number of introns compared to those in Groups B and C. This finding suggests that Group A genes may enhance the evolution of biological functions through mechanisms such as recombination and increased complexity during the evolutionary process [61].”
L455-457: “The specific distribution of developmental and cellular process regulatory elements in the promoter regions of GH18 family genes reveals the potential functions of these genes in the growth and development of T6.” was changed to “The specific distribution of developmental and cellular process regulatory elements within the promoter regions of GH18 family genes suggests their potential roles in the growth and development of T6.”
L491-494: “These findings collectively suggest that GH18-3 and GH18-16 may possess the capability to degrade the cell walls of pathogenic fungi and the cyst walls of nematodes, thereby endowing it with potential dual functions of antimicrobial and nematicidal activities.” was changed to “These findings collectively suggest that GH18-3 and GH18-16 may possess the capability to degrade the cell walls of pathogenic fungi and the cyst walls of nematodes, thereby endowing them with potential dual functions of antimicrobial and nematicidal activities.”

Round 2
Reviewer 1 Report
Thanks to the authors for their efforts to improve the article.
-
Author Response
Dear Reviewer
We sincerely appreciate your efforts in handling the review of our manuscript. We extend our heartfelt thanks for your positive consideration of our manuscript. Thank you very much.
With best wishes,
Yours sincerely,
Cizhong Duan
Reviewer 2 Report
General comments
• In the introduction section, I consider it necessary to provide more information on the other control methods used in pest management and link it to why T. longibrachiatum is better.
• In the methodology section, a characterization of the H. avenae sampling area is needed.
Please see the attachment.

Author Response
Response to Reviewer #2
Dear Reviewer
We would like to express our sincere gratitude for your careful and thorough reading of this manuscript (jof-3841329), as well as for your thoughtful comments and constructive suggestions, which have significantly enhanced its quality. We are deeply appreciative of your selfless and generous assistance. Your insightful feedback has provided invaluable support in the revision process, for which we are immensely thankful. In accordance with your valuable suggestions, we have meticulously revised the manuscript and incorporated your comments into the preparation of the revised version. All amendments are highlighted in red within the revised manuscript. For further details, please refer to the revised manuscript. We extend our heartfelt thanks for your positive consideration of our work.
With best wishes,
Yours sincerely,
Cizhong Duan
Major comment:
- In the introduction section, I consider it necessary to provide more information on the other control methods used in pest management and link it to why T. longibrachiatum is better.
- In the methodology section, a characterization of the H. avenae sampling area is needed.
Response:
Thank you. We would like to thank you for careful and thorough reading of this manuscript and for your thoughtful comments and constructive suggestions, which help to improve the quality of this manuscript. According to your valuable comments, we have made careful revision and incorporated your comments in the preparation of the revised version of our manuscript. More details please see the revised manuscript.
Comment 1:
In the introduction section, I consider it necessary to provide more information on the other control methods used in pest management and link it to why T. longibrachiatum is better.
Response:
We are deeply grateful for your contributions to the improvement of this paper. We greatly appreciate your constructive feedback, which highlights an aspect we had previously overlooked. Indeed, it is essential to provide additional information regarding alternative methods for controlling Heterodera avenae, as this will more effectively emphasize the advantages of Trichoderma longibrachiatum T6. Accordingly, we have made careful and detailed revisions based on your guidance. Please refer to lines 52-57, 57-61, 61-64, 65-67, 67-68, and 68-70 in the revised manuscript. The specific modifications are outlined below:
L52-57: “Traditional agricultural control measures, such as crop rotation [6] and the use of resistant varieties, can reduce the density of H. avenae to a certain extent; however, their implementation is often constrained. This limitation primarily arises from the insufficient economic feasibility associated with long-term crop rotation and the vulnerability of resistant varieties to infection by the second-stage juveniles (J2) of H. avenae during the early growth stages [7].” was added.
L57-61: “In light of the inherent constraints associated with chemical control approaches, particularly the emergence of nematode resistance, environmental contamination, and ecological hazards [6].” was changed to “While chemical nematicides are favored for their low cost and rapid effectiveness and have historically served as the primary strategy for managing H. avenae [22], their sustainable application is hindered by inherent limitations, including the emergence of nematode resistance, environmental pollution, and ecological hazards [23].”
L61-64: “Thus, the development of efficient and environmentally biological control agents has become the key focus in management of H. avenae.” was changed to “Biological control has the potential to efficiently and specifically manage nematodes, significantly reducing reliance on chemical pesticides and mitigating environmental pollution. The development of biological control agents represents a key direction in promoting the advancement of green agriculture.”
L65-67: “Trichoderma spp. are beneficial fungi that recognized for their broad-spectrum activity, safety, adaptability, and diverse mechanisms of action [7-11].” was changed to “Trichoderma spp. are widely recognized as effective biological control agents due to their broad-spectrum activity, safety, adaptability, and diverse mechanisms of action [7-11].”
L67-68: “Current studies have demonstrated that T. longibrachiatum [15], T. viride [16], T. harzianum [17], T. koningiopsis [18], and T. hamatum [19] can effectively control PPNs.” was added.
L68-70: “Zhang et al. [20] found that the T. longibrachiatum T6 (T6) strain exhibited intense parasitic and lethal effects on H. avenae.” was added.
Comment 2:
In the methodology section, a characterization of the H. avenae sampling area is needed.
Response:
We deeply appreciate your contributions to the improvement of this article. Accordingly, we have conducted careful and detailed revisions based on your guidance. We have described the sampling area of H. avenae in accordance with your constructive suggestions. Please refer to lines 136-140 in the revised manuscript. The specific modifications are outlined below:
L125-129: “Soil samples were collected from the rhizosphere and tillage layer at a depth of 5 to 20 cm using the Z-shaped sampling method. These samples were taken from wheat plants exhibiting chlorosis, yellowing, and stunted growth as a result of infection by H. avenae in Suzhou City, Anhui Province, China.” was added.
Detailed comments
Comment 1:
L.11: Add a vernacular name
Response:
We are deeply grateful for your contributions to the improvement of this paper. According to your suggestion, I have added the vernacular name of Heterodera avenae (cereal cyst nematode). Please refer to lines 11 in the revised manuscript. The specific modifications are outlined below:
L11: “Heterodera avenae (Tylenchida: Heteroderidae) causes significant economic losses in the production of wheat, barley, and other cereal crops worldwide.” was changed to “The cereal cyst nematode, Heterodera avenae (Tylenchida: Heteroderidae), is responsible for substantial economic losses in the global production of wheat, barley, and other cereal crops.”
Comment 2:
L.12: “losses in the production of wheat”
Response:
We are deeply grateful for your contributions to the improvement of this paper. In accordance with your suggestion, I have revised this inappropriate expression. Please refer to lines 11-13 in the revised manuscript. The specific modifications are outlined below:
L11-13: “Heterodera avenae (Tylenchida: Heteroderidae) causes significant economic losses in the production of wheat, barley, and other cereal crops worldwide.” was changed to “The cereal cyst nematode, Heterodera avenae (Tylenchida: Heteroderidae), is responsible for substantial economic losses in the global production of wheat, barley, and other cereal crops.”
Comment 3:
L.34: “We need some discussion and conclution statements”
Response:
We are deeply grateful for your contributions to the improvement of this paper. In accordance with your constructive suggestions, I have added the sections that needed to be supplemented. Please refer to lines 32-34, 35-37 and 37-41 in the revised manuscript. The specific modifications are outlined below:
L32-35: “Meanwhile, we found that the tertiary structures of GH18-3 and GH18-16 proteins exhibited the highest homology with Chit33 and Chit42, which are key enzymes for antifungal activity in T. harzianum.” was changed to “Meanwhile, we found that the GH18-3 and GH18-16 proteins exhibit the highest homology with key enzymes responsible for antifungal activity in T. harzianum, demonstrating dual biocontrol potential in both antifungal activity and nematode control.”
L35-37: “Overall, these results indicate that the GH18 family has undergone functional diversification during evolution, with each member assuming specific biological roles in parasitic nematodes.” was added.
L37-41: “This study provides a theoretical foundation for identifying novel nematicidal genes from Trichoderma and for cultivating highly efficient biocontrol strains through transgenic engineering, which holds significant practical implications for advancing the green control of plant-parasitic nematodes (PPNs).” was added.
Comment 4:
L.37: Remove the Keywords "genome-wide analysis".
Response:
We are deeply grateful for your contributions to the improvement of this paper. Based on your suggestion, I have removed this keyword. Please refer to lines 42 in the revised manuscript. The specific modifications are outlined below:
Keywords: biocontrol; glycoside hydrolase 18 chitinase family; expression characteristic; genome-wide analysis
Comment 5:
L.43: “The Heterodera avenae is one of the PPNs that poses serious threat to wheat production [3] and causes substantial economic losses in major wheat-producing regions worldwide (over 40 countries and regions including North Africa, West Asia, India, China, Australia, and Europe) [4].” Remove definite article in this sentence.
Response:
We are deeply grateful for your contributions to the improvement of this paper. We have revised my manuscript according to your suggestion. Please refer to lines 48-51 in the revised manuscript. The specific modifications are outlined below:
L48-51: “The Heterodera avenae is one of the PPNs that poses serious threat to wheat production [3] and causes substantial economic losses in major wheat-producing regions worldwide (over 40 countries and regions including North Africa, West Asia, India, China, Australia, and Europe) [4].” was changed to “Heterodera avenae is one of the PPNs that poses serious threat to wheat production [3] and causes substantial economic losses in major wheat-producing regions worldwide (over 40 countries and regions including North Africa, West Asia, India, China, Australia, and Europe) [4].”
Comment 6:
L.47: “In China, the losses in wheat production that attributed to H. avenae was ranged from 10% to 35% [5].” Remove definite article in this sentence.
Response:
We are deeply grateful for your contributions to the improvement of this paper. We have revised my manuscript according to your suggestion. Please refer to lines 51-52 in the revised manuscript. The specific modifications are outlined below:
L51-52: “In China, the losses in wheat production that attributed to H. avenae was ranged from 10% to 35% [5].” was changed to “In China, losses in wheat production that attributed to H. avenae was ranged from 10% to 35% [5].”
Comment 7:
L.53-54: “Trichoderma spp. are beneficial fungi that recognized for their broad-spectrum activity, safety, adaptability, and diverse mechanisms of action [7-11].” Remove beneficial fungi in this sentence.
Response:
We are deeply grateful for your contributions to the improvement of this paper. We have revised my manuscript according to your suggestion. Please refer to lines 65-67 in the revised manuscript. The specific modifications are outlined below:
L65-67: “Trichoderma spp. are beneficial fungi that recognized for their broad-spectrum activity, safety, adaptability, and diverse mechanisms of action [7-11].” was changed to “Trichoderma spp. are widely recognized as effective biological control agents due to their broad-spectrum activity, safety, adaptability, and diverse mechanisms of action [10-14].”
Comment 8:
L.88: “There is no justification on the research performed. So, we need to state clearly a real problem to study genome-wide... Justify: Provide a contextual background. real needs. Then, Indicate scientific benefits, economic benefits. ”
Response:
We sincerely appreciate your thorough review of our manuscript and your constructive suggestions. We have revised the manuscript in accordance with your comments and have incorporated the necessary content. Please refer to lines 97-106 and 107-115 in the revised manuscript. The relevant changes are as follows:
L97-106: “Current research predominantly focuses on the functional validation of individual genes. In contrast, Yang et al. discovered that genes in the GH18 family of Arthrobotrys oligospora (Orbiliales: Orbiliaceae) exhibit differential expression across various growth stages and demonstrate diverse regulatory functions during growth, differentiation, and pathogen infection processes [31]. Given the complexity of the functional regulation of this gene family, systematic functional identification and analysis of regulatory mechanisms are particularly crucial. However, there is still a lack of systematic research on the T6 GH18 gene family, particularly regarding which members are involved in the transcriptional regulation of nematocidal mechanisms during the parasitism of the H. avenae cysts, which urgently requires in-depth investigation.” was changed to “The growth and morphological development of fungi necessitate cell wall remodeling, a process that requires the coordinated action of various cell wall-related hydrolases, particularly the GH18 family of chitinases [34,35]. These enzymes enhance nutrient utilization by facilitating endogenous cell wall autolysis and degrading exogenous chitin to meet nutritional demands [36-38]. Despite these essential functions, the GH18 family exhibits functional redundancy and significant variation in gene number among Trichoderma spp. [39,40]. Even within the same species, GH18 family genes demonstrate structural and functional domain differentiation. Nevertheless, the precise identification and systematic analysis of GH18 family genes in T6 have not yet been conducted, which limits a deeper understanding of the functions of its chitinases and their potential in biological control.”
L107-115: “In recent years, the overexpression of the key GH18 gene through genetic engineering has successfully led to the development of engineered strains with enhanced antifungal activity [27], significantly improving biocontrol efficacy against pathogenic fungi and reducing the costs associated with biological disease control. However, research on the key functional information of the GH18 family in T6 remains insufficient, particularly regarding which members are involved in transcriptional regulation and contribute to the nematocidal mechanism during the parasitism of H. avenae. This gap in knowledge severely restricts the development of efficient and low-cost biocontrol agents targeting this objective.” was added.
Comment 9:
L.94: “To address these gaps, the genome-wide identification of the physicochemical properties, evolutionary relationships, gene structures, and protein domains of T6 GH18 family genes were systematically analyzed their present study. ”
Response:
We sincerely appreciate your thorough review of our manuscript and your constructive suggestions. I sincerely apologize for any misunderstanding caused by my previous expression. In accordance with your request, I have revised the phrase “To address these gaps”. Please refer to lines 115-117 in the revised manuscript. The relevant changes are as follows:
L115-117: “To address these gaps, the genome-wide identification of the physicochemical properties, evolutionary relationships, gene structures, and protein domains of T6 GH18 family genes were systematically analyzed their present study.” was changed to “Therefore, we systematically characterized the GH18 family genes in T6 through genome-wide identification, including their physicochemical properties, evolutionary relationships, gene structures, and protein domains.”
Comment 10:
The English could be improved to more clearly express the research.
Response:
We are deeply grateful for your contributions to the improvement of this paper. Thank you very much. As per your suggestion, the entire manuscript has been reviewed by Dr. Mohammed Mujitaba Dawuda who is from English speaking country (Department of Horticulture, FoA, University for Development Studies, P. O. Box TL 1882, Tamale, Ghana). He has provided us with many valuable comments and revision suggestions, and we have carefully revised our manuscript based on his recommendations. For more details, please refer to the revised manuscript. Please refer to lines L19-21, L21-22, L22-24, L46-47,L138-143,L146-147,L165-168,L183-185,L185-187,L188-192,L192-193,L205-206,L210-212,L230-231,L236-238,L266-267,L269-270,L270-272,L282,L296-298,L370-372,L375-376,L382-384,L390-392,L396-397,L397-398,L403-405,L407-408,L537-539,L539-542 in the revised manuscript. The relevant changes are as follows:
L19-21: “This analysis encompassed gene structure, evolutionary development, protein characteristics, and gene expression profiles following nematode parasitism, as determined by qRT-PCR.” was changed to “The analysis encompassed gene structure, evolutionary development, protein characteristics, and gene expression profiles following nematode parasitism, as determined by qRT-PCR.”
L21-22: “Our results indicate that 18 GH18 members in T6 were clustered into three major groups (A, B, C), which comprise seven subgroups.” was changed to “Our results indicate that 18 GH18 members in T6 were clustered into three major groups (A, B, and C), which comprise seven subgroups.”
L22-24: “Each subgroup exhibits highly conserved catalytic domains, motifs, and gene structures, while the cis-acting elements show extensive responses to hormones, stress-related signals, and light.” was changed to “Each subgroup exhibits highly conserved catalytic domains, motifs, and gene structures, while the cis-acting elements demonstrate extensive responsiveness to hormones, stress-related signals, and light.”
L46-47: “It is estimated that global agricultural production incurs losses of up to $215.77 billion annually due to PPNs [2].” was changed to “It is estimated that global agricultural production suffers losses of up to $215.77 billion annually due to PPNs [2].”
L138-143: “Additionally, the comprehensive bioinformatics analyses were conducted for identifying the family members, and the parameters of molecular weight, theoretical isoelectric point (pI), instability index, aliphatic index, GRAVY score, subcellular localization, signal peptides, transmembrane topology [32], and phosphorylation sites [33] were analyzed and predicted using the bioinformatics analysis tool that listed in Table S1.” was changed to “Additionally, the comprehensive bioinformatics analysis was conducted for identifying the family members, and the parameters of molecular weight, theoretical isoelectric point (pI), instability index, aliphatic index, GRAVY score, subcellular localization, signal peptides, transmembrane topology [41], and phosphorylation sites [42] were analyzed and predicted using the bioinformatics analysis tool that listed in Table S1.”
L146-147: “Gene locations were visualized using the “Gene Location Visualize from GTF/GFF file” function in TBtools [34].” was changed to “Gene locations were visualized using the “Gene Location Visualize from GTF/GFF file” function within TBtools [43].”
L165-168: “Promoter sequences, specifically those located 2,000 base pairs upstream of the translation start site of T. longibrachiatum T6 family GH18 genes, were extracted from the genome database using the “Gtf/Gff3 Sequences Extract” and “Fasta Extract (Recommended)” functions in TBtools.” was changed to “Promoter sequences, specifically those located 2,000 base pairs upstream of the translation start site of T6 GH18 family genes, were extracted from the genome database using the “Gtf/Gff3 Sequences Extract” and “Fasta Extract (Recommended)” functions within TBtools.”
L183-185: “To analyze the GH18 family genes expression characteristics after the T6 parasitism on the H. avenae cysts, the methods and sample preparation were conducted according to Wang et al. [44] with minimal modifications.” was changed to “To analyze the GH18 family genes expression characteristics after the T6 parasitism on the H. avenae cysts, the methods and samples preparation were conducted according to Wang et al. [53] with minimal modification.”
L185-187: “To analyze the GH18 family genes expression characteristics after the T6 parasitism on the H. avenae cysts, the methods and sample preparation were conducted according to Wang et al. [44] with minimal modifications.” was changed to “To analyze the GH18 family genes expression characteristics after the T6 parasitism on the H. avenae cysts, the methods and samples preparation were conducted according to Wang et al. [53] with minimal modification.”
L188-192: “Total RNA from each sample was converted to cDNA using the RevertAid First Strand cDNA Synthesis Kit (Invitrogen) and then real-time quantitative PCR was performed using the TB Green® Premix Ex Taq™ II kit (Takara, Dalian, China) with Actin (accession number XM_024923451.1) as the internal reference gene.” was changed to “Total RNA from each sample was converted to cDNA using the RevertAid First Strand cDNA Synthesis Kit (Invitrogen) and then the real-time quantitative PCR (qRT-PCR) was performed using the TB Green® Premix Ex Taq™ II kit (Takara, Dalian, China) with Actin (accession number XM_024923451.1) as the internal reference gene.”
L192-193: “Table S2 lists the specific primers for GH18 family genes and the primers for the internal reference gene used for quantitative RT-PCR analysis.” was changed to “Table S2 lists the specific primers for GH18 family genes and the primers for the internal reference gene used for qRT-PCR analysis.”
L205-206: “Ultimately, 18 members were confirmed to belong to the GH18 gene family in T6.” was changed to “Ultimately, 18 members were confirmed to belong to the GH18 family genes in T6.”
L210-212: “Physicochemical property analysis revealed that the GH18 proteins exhibited molecular weights between 34.53 and 159.28 kDa, with theoretical isoelectric points (pI) ranging from 4.20 to 6.20.” was changed to “Physicochemical property analysis revealed that the GH18 proteins exhibited molecular weights from 34.53 to 159.28 kDa, with theoretical isoelectric points (pI) ranging from 4.20 to 6.20.”
L230-231: “The number of GH18 genes on each chromosome was not correlated with chromosome size.” was changed to “The number of GH18 family genes on each chromosome was not correlated with chromosome size.”
L236-238: “Similarly, in T. reesei, T. harzianum, and T. virens, there were 5, 12, and 19 gene duplication events, respectively, with their Ka/Ks ratios also remaining below 1.” was changed to “Similarly , there were 5, 12, and 19 gene duplication events in T. reesei, T. harzianum, and T. virens, respectively, with their Ka/Ks ratios also remaining below 1.”
L266-267: “Furthermore, these proteins were categorized into three major groups (A, B, and C) and eight subgroups (A-II, A-IV, A-V; B-I, B-II, B-V; C-I; C-II) utilizing the maximum likelihood method implemented in MEGA11.” was changed to “Furthermore, these proteins were classified into three major groups (A, B, and C) and eight subgroups (A-II, A-IV, A-V; B-I, B-II, B-V; C-I; C-II).”
L269-270: “Pfam domain analysis indicated that all GH18 proteins contained the Glyco_18 domain, which is characteristic of GH18 chitinases (Figure 5B).” was changed to “Pfam domain analysis indicated that all GH18 proteins contained the Glyco_18 domain, which was characteristic of GH18 chitinases (Figure 5B).”
L270-272: “To characterize the GH18 family proteins, conserved motifs were analyzed using MEME.” was changed to “Remarkably, Group C possessed a unique LysM domain. To characterize the GH18 family proteins, the conserved motifs were analyzed using MEME.”
L282: “Analysis of the gene structures of 18 GH18 family members.” was changed to “Analysis of the gene structure of 18 GH18 family members.”
L296-298: “Stress-responsive elements (encompassing regulatory element anaerobic, low-temperature responsive, defense and stress responsive) were present in 12 members.” was changed to “Stress-responsive elements (encompassing regulatory element anaerobic, low-temperature responsive, defense and stress responsive) were presented in 12 members.”
L370-372: “To investigate the transcriptional responses of GH18 family genes during parasitism with H. avenae cysts, quantitative RT-PCR was performed to assess the relative expression levels of 18 GH18 genes in T6 at various time points post-inoculation (Figure 11).” was changed to “To investigate the transcriptional responses of GH18 family genes during T6 parasitism with H. avenae cysts, qRT-PCR was performed to assess the relative expression levels of 18 GH18 genes in T6 at various time points post-inoculation (Figure 11).”
L375-376: “The relative expression levels of the GH18 family genes at different time points after T6 parasitized on H. avenae cysts.” was changed to “The relative expression levels of the GH18 family genes at different time points after T6 mycelia parasitized on the surface of the H. avenae cysts.”
L382-384: “At 12 h post-inoculation, GH18-1, GH18-3, GH18-4, GH18-8, GH18-10, GH18-11, GH18-15 and GH18-16 were significantly upregulated (P < 0.05) compared to controls.” was changed to “At 12 h after T6 mycelia parasitized on the surface of the cysts, the expression levels of GH18-1, GH18-3, GH18-4, GH18-8, GH18-10, GH18-11, GH18-15, and GH18-16 were significantly upregulated compared to the control group (P < 0.05).”
L390-392: “Notably, the expression levels of the GH18-3 (Group B) and GH18-16 (Group A) at 12, 24, 48, and 72 h were significantly higher compared to the control group than those of other GH18 family genes.” was changed to “Notably, the expression levels of the GH18-3 (Group B) and GH18-16 (Group A) at 12, 24, 48, and 72 h were significantly higher compared to the control group and other GH18 family genes.”
L396-397: “Similarly, the expression level of the GH18-16 gene showed increases of 2.00, 4.29, 4.70, 2.68, and 1.81-fold at the same time points.” was changed to “Similarly, the expression level of the GH18-16 gene was increased by 2.00, 4.29, 4.70, 2.68, and 1.81-fold at the same time points.”
L397-398: “Combined with genomic information analysis and quantitative data, it appears that GH18-3 and GH18-16 play significant roles in the parasitism of H. avenae cysts by T6.” was changed to “Combined with genomic information analysis and quantitative data, it appears that GH18-3 and GH18-16 play significant roles in T6 parasitism of H. avenae cysts.”
L403-405: “Within the genus Trichoderma spp., the GH18 family of chitinases constitutes a diverse multigene family that plays a critical role in essential biological processes, including growth, nutrient acquisition, interspecies interactions, and defense [46,47].” was changed to “Within the genus Trichoderma, the GH18 family of chitinases constitutes a diverse multigene family that plays a critical role in essential biological processes, including growth, nutrient acquisition, interspecies interactions, and defense [55,56].”
L407-408: “In this study, a genome-wide analysis identified 18 GH18 family genes in the T6, consistent with the gene family size reported in T. reesei [52].” was changed to “In this study, a genome-wide analysis identified 18 GH18 family genes in T6, consistent with T. reesei [40].”
L537-539: “Crucially, GH18-3 and GH18-16 were identified as core genes interacting with nematodes, providing essential targets for in-depth exploration of the molecular mechanisms by which Trichoderma spp. chitinases antagonize H. avenae parasitism.” was changed to “Crucially, GH18-3 and GH18-16 were identified as key genes interacting with nematodes, providing essential targets for in-depth exploration of the molecular mechanisms by which Trichoderma spp. chitinases parasitism on H. avenae.”
L539-542: “The findings of this study not only deepen the fundamental theoretical research on chitinase-based biopesticides but also provide an external impetus for the genetic engineering modification of strains, thereby promoting the advancement of Trichoderma field control technologies.” was changed to “The findings of this study not only enhance the fundamental theoretical research on chitinase-based biopesticides but also provide an external impetus for the genetic engineering modification of strains, thereby promoting the advancement of Trichoderma field control technologies.”